# A joint view on genetic variants for adiposity differentiates subtypes with distinct metabolic implications

Thomas W Winkler [1], Felix Günther[1,2], Simon Höllerer[1], Martina Zimmermann [1], Ruth JF Loos [3,4,5], Zoltán Kutalik[6,7] & Iris M Heid[1]

The problem of the genetics of related phenotypes is often addressed by analyzing adjusted-model traits, but such traits warrant cautious interpretation. Here, we adopt a joint view of adiposity traits in ~322,154 subjects (GIANT consortium). We classify 159 signals associated with body mass index (BMI), waist-to-hip ratio (WHR), or WHR adjusted for BMI (WHRadjBMI) at $P < 5 \times 10^{-8}$, into four classes based on the direction of their effects on BMI and WHR. Our classes help differentiate adiposity genetics with respect to anthropometry, fat depots, and metabolic health. Class-specific Mendelian randomization reveals that variants associated with both WHR-decrease and BMI increase are linked to metabolically rather favorable adiposity through beneficial hip fat. Class-specific enrichment analyses implicate digestive systems as a pathway in adiposity genetics. Our results demonstrate that WHRadjBMI variants capture relevant effects of "unexpected fat distribution given the BMI" and that a joint view of the genetics underlying related phenotypes can inform on important biology.

[1] Department of Genetic Epidemiology, University of Regensburg, D-93051 Regensburg, Germany. [2] Statistical Consulting Unit StaBLab, Department of Statistics, Ludwig-Maximilians-Universität Munich, D-80539 Munich, Germany. [3] The Charles Bronfman Institute for Personalized Medicine, Icahn School of Medicine at Mount Sinai, 10029 New York, NY, USA. [4] The Genetics of Obesity and Related Metabolic Traits Program, Icahn School of Medicine at Mount Sinai, 10029 New York, NY, USA. [5] The Mindich Child health and Development Institute, Icahn School of Medicine at Mount Sinai, 10029 New York, NY, USA. [6] Institute of Social and Preventive Medicine (IUMSP), Centre Hospitalier Universitaire Vaudois (CHUV), 1010 Lausanne, Switzerland. [7] Swiss Institute of Bioinformatics, 1015 Lausanne, Switzerland. These authors jointly supervised this work: Zoltán Kutalik, Iris M Heid. Correspondence and requests for materials should be addressed to T.W.W. (email: thomas.winkler@ukr.de) or to I.M.H. (email: iris.heid@ukr.de)

G enome-wide association studies (GWAS) have become a well-established and very successful approach to understand the genetic background of disease phenotypes. However, for our understanding of the underlying mechanisms, it is an important challenge to disentangle the genetics of related phenotypes. Frequently, this is approached by using an adjusted-model trait where the trait Y is adjusted for a covariate Z (YadjZ) in order to separate the genetics of YadjZ from the genetics of Z. However, these adjusted-model traits warrant cautious interpretation: as Aschard and colleagues pointed out, genome scans for traits adjusted for heritable covariates reveal not only genetic factors for the phenotype Y, but also those of the covariate Z to an extent that depends on their correlation[1].

We exemplify this issue on adiposity traits that were also utilized by Aschard and colleagues[1]: BMI and WHR are correlated and capture two aspects of adiposity, overall fat mass, and fat distribution, respectively. Both are independently associated with type 2 diabetes (T2D), coronary artery disease (CAD), and mortality[2]. The phenotypic correlation between BMI and WHR and the biological mechanisms linking these two measures hamper the distinction of their genetic make-up[3,4]. Recently, meta-analyses by the GIANT consortium highlighted hundreds of associated loci for BMI, WHR, and WHRadjBMI[5,6], whereas BMI and WHRadjBMI loci were shown to depict different biological processes (neuronal versus metabolic), a direct comparison of the loci was lacking.

Aschard and colleagues pointed out that some of the WHRadjBMI lead variants were not completely independent of BMI and showed some effect on BMI in the unexpected direction (WHR increasing allele decreased BMI). This is due to the fact that the genetic effect estimate for WHRadjBMI, $b_{\mathrm{WHRadjBMI}}$, is related to the estimate for WHR, $b_{\mathrm{WHR}}$, and the estimate for BMI, $b_{\mathrm{BMI}}$ by $b_{\mathrm{WHRadjBMI}} = b_{\mathrm{WHR}} - r \ast b_{\mathrm{BMI}}$, with $r$ being the observational correlation between BMI and WHR in the analyzed study[1]. A genome-wide scan on WHRadjBMI will thus not only identify genetic factors for WHR, but will also tend to pick up variants with an additional opposite effect on BMI, or even an effect on BMI only when the sample size is large enough. Aschard and colleagues extended their point by cautioning against potentially false positive signals and biased genetic effect estimates. They propose to examine the potential of the bias by investigating the corrected effect $b_{\mathrm{WHRadjBMI}} + r \ast b_{\mathrm{BMI}}$ to ensure that an established WHRadjBMI-association is not biased by the BMI-association.

Therefore, the genome screening of adjusted-model traits in general, and WHRadjBMI in particular, has been criticized[1] for its potential to yield biased estimates and spurious associations. As a consequence, it is a current concern whether adjusted-model trait loci like WHRadjBMI loci can reveal meaningful biological information or whether they represent uninterpretable artefacts. We thus investigated the genetic variants that are associated genome-wide with BMI, WHR, or WHRadjBMI in the GIANT data[5,6] with regard to their co-association with BMI and WHR and the link of this co-association to metabolic health and pathways. We find that the joint view of the genetic variants across all three of the adiposity traits helps differentiate adiposity subtypes with distinct fat depots and distinct metabolic implications. Furthermore, the joint view helps resolve some of the issues that derive from conducting GWAS on adjusted-model traits.

## Results

**Little contribution of the WHR genomic screen.** In order to define a set of adiposity-associated variants as the basis of our investigation, we selected variants that showed European ancestry based genome-wide significant association ($P < 5 \times 10^{-8}$) with

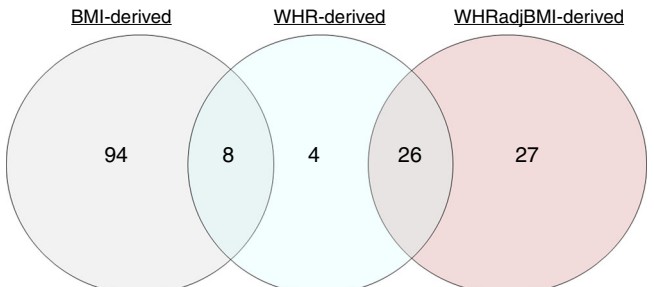

**Fig. 1** Identification of 159 signals from three genomic scans. The Venn diagram shows the number of independent genome-wide significant ($P < 5 \times 10^{-8}$) signals derived from the BMI−, the WHR−, or the WHRadjBMI-scan, respectively, and their overlap. We found no overlap between BMI- and WHRadjBMI-derived variants

any of the three adiposity traits, BMI, WHR, and WHRadjBMI from the GIANT consortium (up to $N = 322{,}135$, see Methods)[5,6]. This yielded 159 independent lead variants ($> 500$kB or $r^2 < 0.1$): 102, 38, or 53 variants genome-wide significant for BMI, WHR, or WHRadjBMI, respectively. We found a substantial overlap of WHR-derived variants (i.e., variants that are genome-wide significant for WHR) with BMI- or WHRadjBMI-derived variants (genome-wide significant for BMI or WHRadjBMI, respectively), with four being exclusive to the WHR-scan, but no overlap between BMI- and WHRadjBMI-derived variants (Fig. 1, Supplementary Data 1). Thus, the WHRadjBMI-derived variants contributed independently from BMI-derived variants in the GIANT data, whereas the WHR-derived variants contributed little beyond.

**WHRadjBMI captures relevant aspects of fat distribution.** Whether or not a genetic variant has "an expected effect on WHR given the BMI effect" (i.e., as expected by the phenotypic correlation $r$, e.g. $r = 0.5$ in the population-based CoLaus study[7]) or "an unexpected effect" can be determined by evaluating the variant's co-association with BMI and WHR: the co-association of the 159 variants is visualized in a plane spanned by the genetic effects on WHR and BMI, i.e., $b_{\mathrm{WHR}}$ vs. $b_{\mathrm{BMI}}$ (Fig. 2a). Variants with a null effect on WHRadjBMI are those with an observed WHR effect to the extent and direction as expected given the variant's BMI effect and the phenotypic correlation $r$ (located on the line $b_{\mathrm{WHR}} = r \ast b_{\mathrm{BMI}}$, with $r = 0.5$, gray dashed line); this is in line with a notion of "an expected change in fat distribution given the change in BMI". Variants with a non-null effect on WHRadjBMI will be those distant from the WHRadjBMI null line. This includes variants with a WHR effect but no effect on BMI, variants with a WHR effect into the opposite direction as their BMI effect, variants with effects on WHR larger than expected from the BMI effect ("supra-expected"), or even BMI effects with no effect on WHR. All are in line with a notion that the observed WHR effect is unexpected given the variant's BMI effect. We hypothesized important insights from a detailed view of these variants' position on the $b_{\mathrm{WHR}}-b_{\mathrm{BMI}}$-plane and the link of this position to physiology and pathology.

**Classifying the 159 adiposity variants.** We classified the 159 variants according to their location on the $b_{\mathrm{WHR}}-b_{\mathrm{BMI}}$-plane (Fig. 2a). We considered an effect as a non-null effect for BMI or WHR, when the effect was nominally significantly different from zero ($P_{\mathrm{BMI}} < 0.05$, $P_{\mathrm{WHR}} < 0.05$, respectively), corresponding to an uncertainty of beta-estimates given by a 95% confidence interval, and as a null effect otherwise. We defined the following four classes: (1) BMI and WHR effects in the same direction ($P_{\mathrm{BMI}} < 0.05$, $P_{\mathrm{WHR}}$

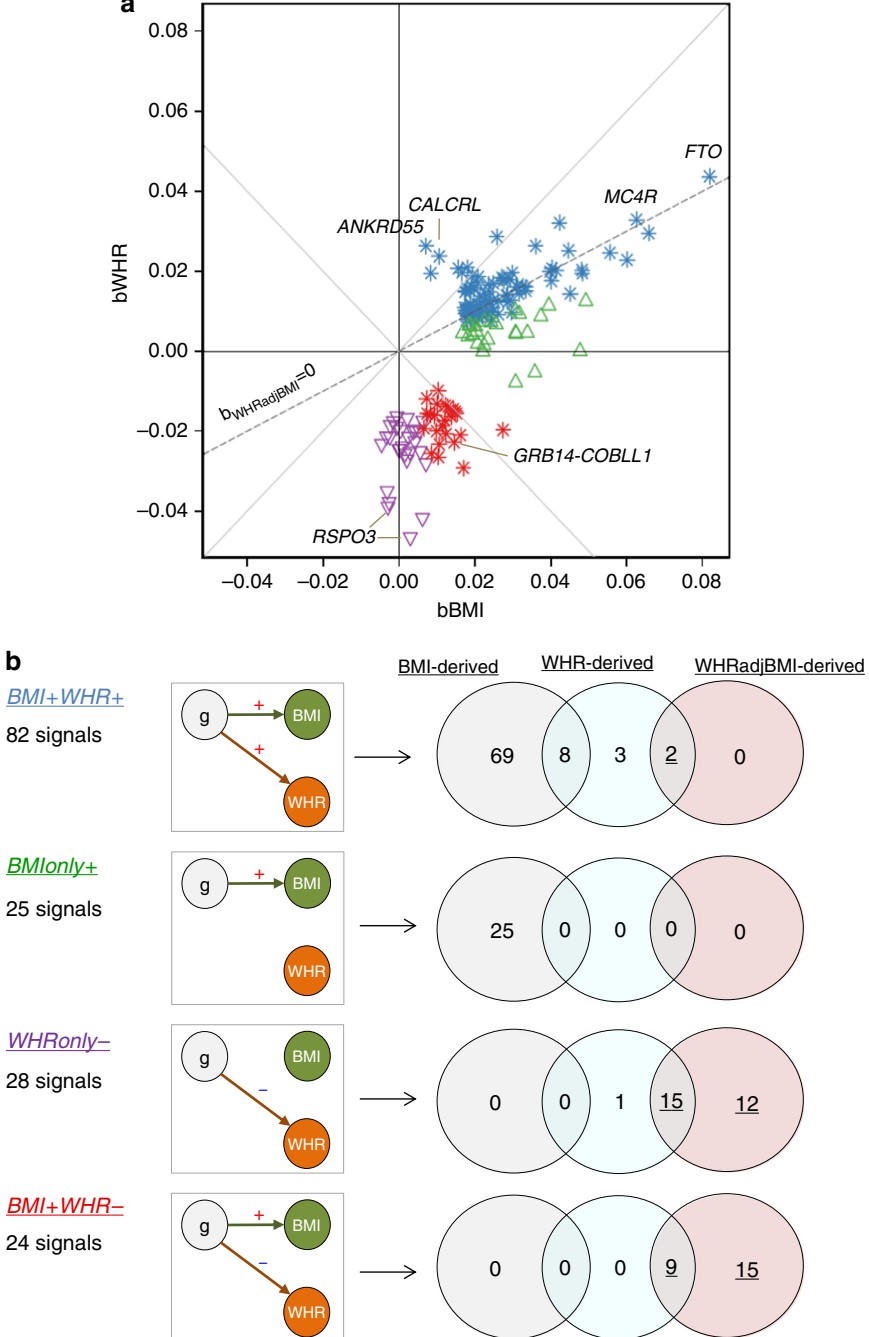

**Fig. 2** Classification of 159 signals and overlap by scan. The figure visualizes the classification of the 159 independent signals according to the position on the $b_{WHR}$-$b_{BMI}$-plane and their overlap by scan. **a** The Scatter plot shows the 159 variants on the $b_{WHR}$-$b_{BMI}$-plane, where $b_{WHR}$ and $b_{BMI}$ are the variant's effect on WHR and BMI, respectively. Coloring indicates the four classes: $BMI + WHR +$ (blue, nominal significant effects on BMI and WHR with consistent directions), $BMIonly +$ (green, nominal significant effects on BMI only), $WHRonly -$ (purple, nominal significant effects on WHR only) and $BMI + WHR -$ (red, nominal significant effects on BMI and WHR with opposite directions). Symbols indicate a nominal significance purely for BMI ($P_{BMI} < 0.05$, $P_{WHR} \geq 0.05$, upward triangle), purely for WHR ($P_{BMI} \geq 0.05$, $P_{WHR} < 0.05$, downward triangle), or for both ($P_{BMI} < 0.05$, $P_{WHR} < 0.05$, stars). The dashed line indicates a null effect for WHRadjBMI ($b_{WHRadjBMI} = 0$, estimated as $b_{WHR} = r*b_{BMI}$, with the correlation between BMI and WHR estimated from the population-based CoLaus study, $r = 0.50$). **b** The diagram shows the number of identified signals per class, illustrates the four classes in directed acyclic graphs and shows Venn diagrams per class to distinguish whether the signals were derived with genome-wide-significance by the BMI-, the WHR- or the WHRadjBMI-scan, or by multiple scans. The underlined numbers reflect the 53 genome-wide significant signals identified by the WHRadjBMI-scan

$< 0.05$; $BMI + WHR +$ ), (2) BMI only effects ($P_{BMI} < 0.05$, $P_{WHR} \geq 0.05$; $BMIonly +$ ), (3) WHR only effects ($P_{WHR} < 0.05$, $P_{BMI} \geq 0.05$; $WHRonly -$ ), (4) BMI and WHR effects into opposite directions ($P_{BMI} < 0.05$, $P_{WHR} < 0.05$; $BMI + WHR -$ ). Of note, the WHR effects that were directionally consistent with the BMI effect, but

larger than expected ("supra-expected") were classified as $BMI + WHR +$ . This classification resulted in 82, 25, 28, or 24 variants for each of the four classes, respectively (Fig. 2a,b).

We found the following: (i) of the 159 variants, the 53 WHRadjBMI-derived variants were all in the $BMI + WHR -$ or

*WHRonly*− class (Fig. 2b, Supplementary Fig. 1), except two variants near *ANKRD55* and *CALCRL* with supra-expected WHR effect (*BMI* + *WHR* + class). All 53 WHRadjBMI-derived variants were orthogonally distant from the WHRadjBMI null line and can be considered effects of "unexpected change in fat distribution given the effect on BMI". (ii) The 102 BMI-derived variants were all in the *BMI* + *WHR* + or *BMIonly* + class (Fig. 2b, Supplementary Fig. 1). They scattered closely around the WHRadjBMI null line with some exceptions in the *BMIonly* + class and are thus, mostly, in line with a notion of a change in fat distribution that is expected given the effect on BMI. (iii) The 38 WHR-derived variants were spread across the classes *BMI* + *WHR* +, *WHRonly*−, or *BMI* + *WHR*− (Fig. 2b, Supplementary Fig. 1); the four variants exclusively identified by the WHR-scan were *BMI* + *WHR* + or *WHRonly*−.

We made further important observations regarding the WHRadjBMI-derived variants: (iv) All 53 WHRadjBMI-derived variants had nominally significant effects on WHR ($P_{WHR} < 0.05$, i.e., no spurious associations, weakest WHR association observed in GIANT $P_{WHR} = 7.5 \times 10^{-3}$, Supplementary Data 1). (v) Of the 53 WHRadjBMI-derived variants, 27 had no effect on BMI ($P_{BMI} \geq 0.05$), 24 had a nominally significant effect on BMI ($P_{BMI} < 0.05$) into the opposite direction. Therefore, WHRadjBMI-derived variants cannot be considered as "independent of BMI".

We conducted two types of sensitivity analyses. First, we re-classified the variants based on different *P*-value thresholds instead of the nominal significance level (Supplementary Data 1). A more stringent threshold at $P < 3 \times 10^{-4}$ ($= 0.05/159$, Bonferroni-corrected) resulted in 36 of the 53 WHRadjBMI-derived variants retaining the class, 11 variants changing from *BMI* + *WHR*− to *WHRonly*−, and six just missing the $P_{WHR} < 3 \times 10^{-4}$ in the GIANT data (one with BMI effect $P_{BMI} < 3 \times 10^{-4}$, five without any effect). However, these six variants showed a significant association with WHR in the independent UK Biobank data ($P_{WHR} < 3 \times 10^{-4}$, $N = 336,107$, $P_{WHR}$ ranging from $9.95 \times 10^{-21}$ to $6.29 \times 10^{-6}$, Supplementary Data 2). Of note, all 53 WHRadjBMI-derived variants showed a significant WHR association in the UK Biobank data ($P_{WHR} < 3 \times 10^{-4}$, Supplementary Data 2). This supports the notion that none of the WHRadjBMI-derived variants from the GIANT data was a spurious association without effect on WHR.

Second, as WHRadjBMI is known for sexually dimorphic genetic effects[8,9], we also conducted a sensitivity analysis re-classifying the 53 WHRadjBMI variants based on their sex-specific effects on WHR and BMI (i.e., women-specific or men-specific classification). Among those, 11 variants showed significant sex-difference in the genetic effect on WHRadjBMI in our data ($P_{Sexdiff} < 0.05/53$). Among those, the 10 variants with women-specific effects retained class in the women-specific, but not in the men-specific classification; similarly, the one variant with men-specific effect retained class in the men-specific, but not in the women-specific classification. For all other variants there was no remarkable pattern by the re-classification for sex-specific effects (Supplementary Data 3).

Generally, with a few exceptions, our classification resulted in splitting the BMI-derived loci into two groups (*BMI* + *WHR* +, *BMIonly* +), and splitting the WHRadjBMI-derived loci into two groups (*BMI* + *WHR*−, *WHRonly*−).

**Computing WHR effect from observed BMI and WHRadjBMI effects**. When $b_{WHRadjBMI}$ and $b_{BMI}$ are given for a variant, $b_{WHR}$ can be computed as $b_{WHRadjBMI} + r^*b_{BMI}$ (or $b_{WHRadjBMI}$ as $b_{WHR} - r^*b_{BMI}$). We aimed to provide empirical data of how good this computation works by comparing the $b_{WHR}$ estimates computed as described above with the observed $b_{WHR}$ (Fig. 3). When

conducting this comparison in one study where we could estimate $r$ directly (interim UK Biobank, $N = 116,295$, r = 0.44), we found perfect agreement between computed and observed $b_{WHR}$ (Spearman correlation coefficient = 0.98). When conducting this comparison in a meta-analysis setting where $r$ could not be estimated directly (i.e., in GIANT, using $r$ from UK Biobank as a reasonable average across GIANT studies), we found still a strong agreement (Spearman correlation coefficient = 0.88). We were able to improve this agreement even further by using sex-stratified correlation estimates (from UK Biobank, $r = 0.46$ for women, 0.60 for men, Spearman correlation coefficient > 0.99) and sex-stratified effect estimates (from GIANT, Spearman correlation coefficient = 0.95). Therefore, the formula $b_{WHR} = b_{WHRadjBMI} + r^*b_{BMI}$ can very well be used to compute unadjusted estimates from adjusted estimates and BMI estimates; the corresponding standard errors are, however, slightly increased yielding lower power (Supplementary Note 1, Supplementary Fig. 2). As a consequence, for consortia working with obesity traits, such as GIANT[5,6], the number of genome-wide traits to be modeled can be limited to two traits as the effect estimate from the third trait can be re-computed with a small loss in precision.

**Anthropometry, fat depots, and cardio-metabolic health**. We were interested in whether the four classes characterized meaningful phenotypes with regard to anthropometry, fat depots, and cardio-metabolic health. We thus derived genetic effects of our 159 variants for such measures from genetic consortia and UK Biobank (see Methods, Supplementary Data 4–7). Effects were aligned for BMI-increasing alleles, where appropriate, and for WHR-decreasing alleles for *WHRonly*− consistent with *BMI* + *WHR*− (resulting in an alignment for hip-increasing alleles in all four classes).

First, when evaluating the 159 variants' co-associations on the components of WHR and BMI, waist and hip circumference, weight, and height (GIANT data, up to $N = 253,239$), we found a clear separation of the four classes (Fig. 4a–b, Supplementary Data 4). This was supported by enrichment and meta-regression based genetic risk score (GRS) analyses ($P_{Binomial} < 3.0 \times 10^{-4}$, Table 1, $P_{GRS} < 8.3 \times 10^{-4}$, Supplementary Table 1, see Methods). Thus, the variants' two-dimensional co-association with BMI and WHR effectively summarizes the 2 × 2 co-associations on (height, weight) and (waist circumference, hip circumference). The class-specific view on the variants' co-association on hip and waist circumference revealed that *BMI* + *WHR* + and *BMIonly* + variants were hip and waist-increasing, *WHRonly*− variants were enriched for hip increase and waist decrease, and the *BMI* + *WHR*− variants were enriched for hip-increasing effects that lacked effects on waist circumference (Table 1). Our results underscore the dual cause for WHR-decreasing effects: decreased waist or increased hip circumference—the role of hip being missed when focusing on "central adiposity" (Supplementary Fig. 3–4; Supplementary Note 2).

Second, we were interested in the variants' impact on more elaborate measures of fat depots including centrally stored visceral adipose tissue (VAT), subcutaneous adipose tissue (SAT) that is ubiquitously stored with a preference at hip and thigh[10–12], and pericardial adipose tissue (PAT), which is a VAT-type fat stored in/around the heart[13]. We evaluated the 159 genetic variants' association on measures derived by bioelectrical impedance (body fat, trunk fat, leg fat; UK Biobank, $N$ up to 114,367) or imaging techniques (SAT, VAT, PAT, VAT/SAT ratio; Ectopic Fat Traits consortium[14], $N$ up to 18,312; Supplementary Data 5, 6). The visualization of the co-association of VAT and SAT was less conclusive (Fig. 4c), whereas enrichment and GRS analyses elucidated a distinct

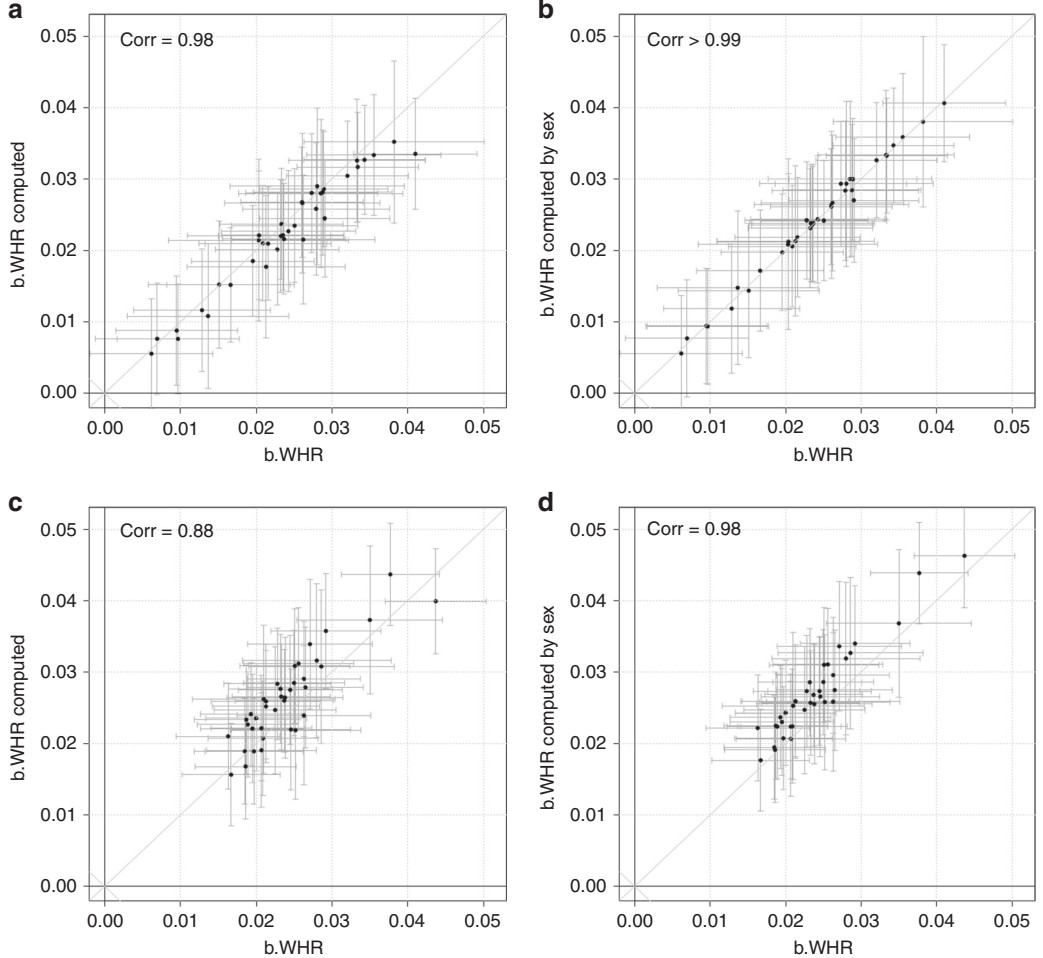

**Fig. 3** Comparison of estimated and computed WHR effect sizes. The figure shows a comparison of effect sizes and standard errors for the 38 genome-wide significant WHR-derived lead variants. Using data from the UK Biobank (UKBB, N = 116,295) as a single large study, we compare estimated overall (sex-combined) WHR effects in UKBB data with **a** computed WHR effects that were calculated from overall BMI and WHRadjBMI effects in UKBB using the overall correlation between WHR and BMI ($r = 0.44$, in UKBB); and with **b** WHR effects that were obtained from meta-analysis of computed sex-specific WHR effects that were calculated from sex-specific BMI and WHRadjBMI effects in UKBB using sex-specific correlations ($r_M = 0.60$, $r_F = 0.46$ in UKBB). Using GIANT meta-analysis summary statistics, we compare meta-analyzed overall WHR effects (resulting from meta-analysis of multiple studies) with **c** computed WHR effects that were calculated from meta-analyzed overall BMI and WHRadjBMI effects using the overall correlation between WHR and BMI ($r = 0.44$, in UKBB), and with **d** WHR effects that were obtained from meta-analysis of computed sex-specific WHR effects that were calculated from meta-analyzed sex-specific BMI and WHRadjBMI effects using sex-specific correlations ($r_M = 0.60$, $r_F = 0.46$ in UKBB)

pattern by class ($P_{Binomial} < 3.0 \times 10^{-4}$, $P_{GRS} < 8.3 \times 10^{-4}$, Table 1, Supplementary Table 1) linking $BMI + WHR+$ to VAT and SAT, $BMIonly-$ and $BMI + WHR-$ only to SAT, and $WHRonly-$ to VAT/SAT ratio.

Third, we evaluated the effects of the 159 variants on eight cardio-metabolic traits (DIAGRAM[15], GLGC[16], MAGIC[17], CARDIoGRAMplusC4D[18], up to $N = 187,135$, Supplementary Data 7). The co-associations on T2D and CAD (Fig. 4d) showed a clear pattern for increasing or decreasing disease risk for the two "extreme" classes $BMI + WHR+$ or $BMI + WHR-$, respectively, but a rather neutral or inconclusive pattern for $BMIonly+$ (except for the known extreme disease effect of $TCF7L2$ into the opposite direction as expected by the BMI effect) and $WHRonly-$. This was supported by enrichment and GRS analyses ($P_{Binomial} < 3.0 \times 10^{-4}$, Table 1, $P_{GRS} < 8.3 \times 10^{-4}$, Supplementary Table 1). The joint impact of the class-specific variants on T2D and CAD was substantial and markedly different: the joint $BMI + WHR+$ alleles increased T2D or CAD risk 2.5- or 1.5-fold, respectively; the joint $BMI + WHR-$ alleles decreased T2D risk to a relative risk of 0.10 and CAD risk to 0.43 (Fig. 5). We found a

consistent pattern for fasting insulin, triglycerides, and HDL-cholesterol (HDL-C, $BMI + WHR+$: adverse, $BMI + WHR-$: protective, Fig. 4e, Table 1, Supplementary Table 1). Overall, the four classes differentiate genetic adiposity effects into metabolically unfavorable ($BMI + WHR+$), metabolically neutral or inconclusive (BMI only, WHR only), and metabolically rather favorable adiposity ($BMI + WHR-$) with some exceptions.

**Evidence of gene expression in digestive system tissue.** Finally, we explored whether our four classes distinguished the underlying physiological pathways. For this, we used DEPICT[19] to search for enriched pathways among the genes overlapping association signals ($P < 10^{-5}$ for any of BMI, WHR, or WHRadjBMI, excluding metabochip data as done previously[5,6], to avoid enriching for known metabolic regions by chip design, see Methods). We applied Data-Driven Expression Prioritized Integration for Complex Traits (DEPICT) for different sets of variants: (i) by the scan that a variant was selected for or (ii) by class. Our scan-specific DEPICT analyses replicated previous

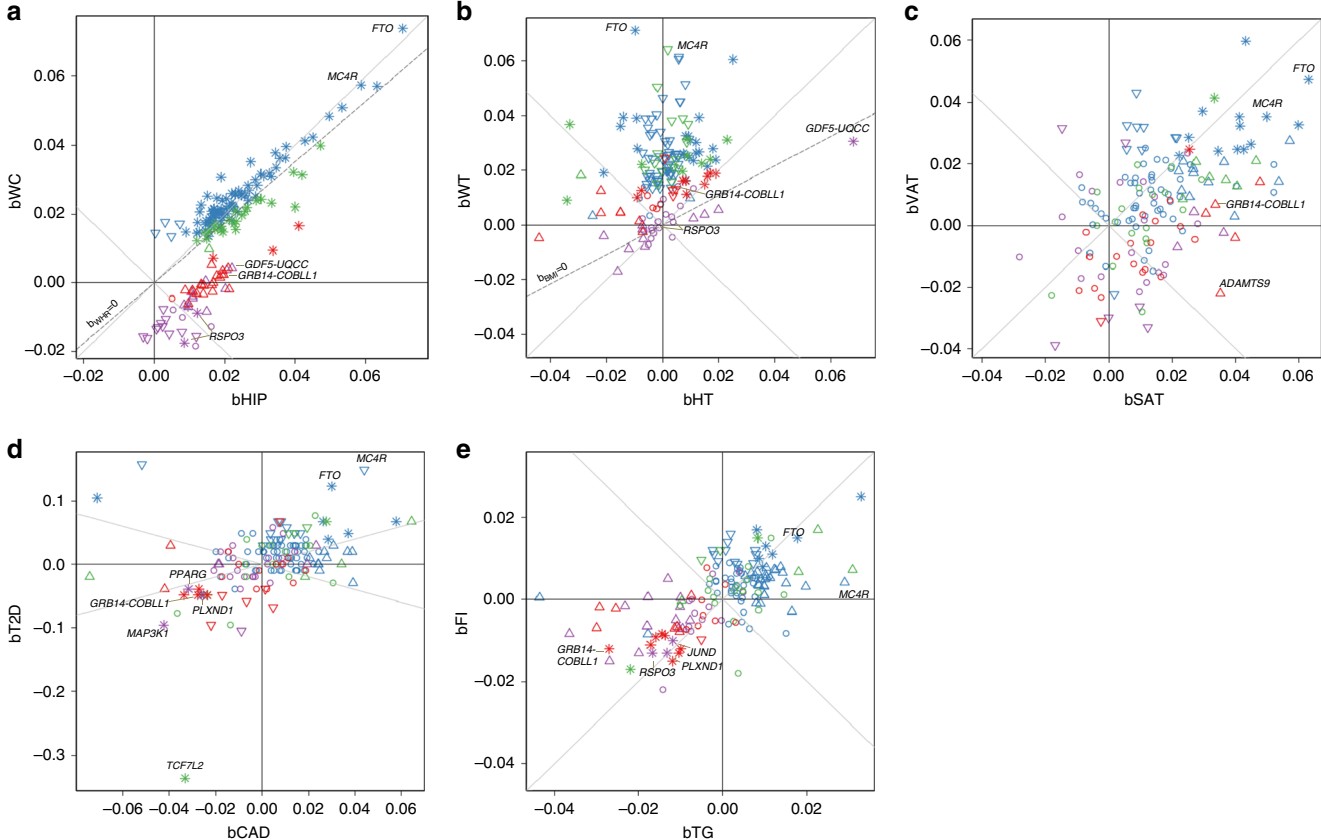

**Fig. 4** Anthropometry, fat depots and cardio-metabolic traits. The figure shows the co-associations for the 159 variants for pairs of traits (from GIANT, DIAGRAM[15], GLGC[16], MAGIC[17], CARDIoGRAMplusC4D[18]): **a** waist circumference (WC) and hip circumference (HIP), **b** weight (WT) and height (HT), **c** visceral adipose tissues (VAT) and subcutaneous adipose tissue (SAT), **d** type 2 diabetes (T2D) and coronary artery disease (CAD), **e** fasting insulin (FI) and triglycerides (TG). Coloring indicates the four classes: $BMI + WHR +$ (blue), $BMIonly +$ (green), $WHRonly-$ (purple), and $BMI + WHR-$ (red). Symbols indicate nominal significance for $y$ axis trait only (downward triangle), the $x$ axis trait only (upward triangle), or both (stars). In **a**, the dashed line indicates a null effect for WHR (slope estimated as $b_{WC}/b_{HIP} = $ mean(WHR), mean(WHR) = 0.88 from CoLaus); in **b**, the dashed line indicates the null effect for BMI (slope estimated as $b_{WT}/b_{HT} = 2*$mean(height)*mean(BMI)*SD(height)/SD(WT) = 0.54, using estimates from CoLaus)

findings[5,6] (highlighting central nervous system, CNS, for BMI-derived variants and adipose tissue for WHRadjBMI). Previous work had not investigated the WHR-derived variants and we found here that they provided an inconclusive pattern without any significant pathway enrichment (judged at false-discovery rate, FDR, < 5%, Supplementary Fig. 5, Supplementary Data 8, Supplementary Note 3). The lack of enriched pathways for WHR-loci suggests that WHR signals capture less-distinct biology than WHRadjBMI or BMI.

Our class-specific DEPICT analyses yielded a pattern for CNS and adipose tissue that was similar to the pattern observed previously by Locke et al. and Shungin et al. for three of our four classes[5,6] (Supplementary Fig. 6, Supplementary Data 9). WHRonly− variants were not only significantly enriched (at FDR < 5%) for adipocyte-related cells and tissues as reported previously[6], but also in physiological systems labeled 'digestive' (rectum, cecum, upper GI, esophagus, stomach) and 'urogenital' (genitals, uterus, endometrium, myometrium) (Fig. 6a, Supplementary Data 9). This WHRonly- class finding was robust, even more pronounced, after excluding known height loci (to remove effects of the known strong height locus around GDF5 and other height regions), after excluding all five RSPO3 signals (to limit the strong contribution of multiple RSPO3 signals in this class), or after using a wider locus definition treating the RSPO3 signals as a single region in the DEPICT analyses (to limit the contribution of

multiple signals like RSPO3, Supplementary Fig. 7, Supplementary Data 10-12).

To follow-up this finding, we used FUMA[20] to examine data from GTEx[21] for tissue-specific enrichments of expression effects of genes overlapping our association results ($P < 10^{-5}$ for any of BMI, WHR, or WHRadjBMI, excluding metabochip data), again separating the variants by class. Consistent with the class-specific DEPICT analyses, genes harboring WHRonly− variants were significantly enriched (Bonferroni-adjusted $P < 0.05$) for expression effects in an adipocyte-related tissue ('Adipose_Subcutaneous') as well as in digestive tissues ('Colon_Sigmoid' and 'Esophagus_Gastroesophageal_Junction', Fig. 6b, Supplementary Data 13, Supplementary Fig. 8). In contrast to DEPICT analyses, there was no significant enrichment for expression effects in urogenital tissue in FUMA analyses; there was an additional significant finding for 'tibial nerve' in FUnctional Mapping and Annotation (FUMA), which is a tissue not included in DEPICT. We found an overlap of nine genes (BARX1, FOXP2, HOXA13, LAMB1, PCK1, PPARG, RGMA, RSPO3, and VEGFA) that contributed to the significant digestive system results in both DEPICT and FUMA tissue-specificity analyses of WHRonly− class variants.

In summary, we identified the digestive system as a pathway for obesity genetics, which highlights an important biology underlying the WHRonly− class variants.

**Table 1 Results of class-specific enrichment analyses**

| Trait | N | BMI + WHR+ n_Tested | Increasing n | P_Binomial | Decreasing n | P_Binomial | BMIonly+ n_Tested | Increasing n | P_Binomial | Decreasing n | P_Binomial | WHRonly− n_Tested | Increasing n | P_Binomial | Decreasing n | P_Binomial | BMI + WHR− n_Tested | Increasing n | P_Binomial | Decreasing n | P_Binomial |
|---|---|---|---|---|---|---|---|---|---|---|---|---|---|---|---|---|---|---|---|---|---|
| Anthropometric traits | | | | | | | | | | | | | | | | | | | | | |
| BMI | 322,135 | 82 | 82 | **4.3E-132** | 0 | 1 | 25 | 25 | **8.9E-41** | 0 | 1 | 28 | 0 | 1 | 0 | 1 | 24 | 24 | **3.6E-39** | 0 | 1 |
| WHR | 212,216 | 82 | 82 | **4.3E-132** | 0 | 1 | 25 | 0 | 1 | 0 | 1 | 28 | 0 | 1 | 28 | **1.4E-45** | 24 | 0 | 1 | 24 | **3.6E-39** |
| WC | 232,083 | 82 | 82 | **4.3E-132** | 0 | 1 | 25 | 24 | **8.7E-38** | 0 | 1 | 28 | 0 | 1 | 14 | **1.1E-15** | 24 | 3 | 0.02 | 0 | 1 |
| HIP | 213,028 | 82 | 78 | **1.7E-119** | 0 | 1 | 25 | 25 | **8.9E-41** | 0 | 1 | 28 | 13 | **3.9E-14** | 0 | 1 | 24 | 23 | **3.3E-36** | 0 | 1 |
| HT | 253,239 | 82 | 17 | **2.0E-11** | 7 | 4.5E-03 | 25 | 6 | **2.9E-05** | 4 | 3.2E-03 | 28 | 4 | 4.9E-03 | 7 | **4.5E-06** | 24 | 7 | **1.5E-06** | 9 | **3.5E-09** |
| WT | 125,943 | 82 | 77 | **1.1E-116** | 0 | 1 | 25 | 24 | **8.7E-38** | 0 | 1 | 28 | 1 | 0.51 | 0 | 1 | 24 | 13 | **2.9E-15** | 0 | 1 |
| Impedance measures | | | | | | | | | | | | | | | | | | | | | |
| Body fat | 114,178 | 80 | 72 | **1.1E-105** | 0 | 1 | 25 | 19 | **5.6E-26** | 1 | 0.47 | 28 | 4 | 4.9E-03 | 2 | 0.15 | 24 | 11 | **4.4E-12** | 0 | 1 |
| Trunk fat | 114,305 | 80 | 72 | **1.1E-105** | 0 | 1 | 25 | 17 | **5.2E-22** | 1 | 0.47 | 28 | 4 | 4.9E-03 | 2 | 0.15 | 24 | 10 | **1.4E-10** | 0 | 1 |
| Leg fat | 114,367 | 80 | 71 | **3.3E-103** | 0 | 1 | 25 | 20 | **4.3E-28** | 0 | 1 | 28 | 2 | 0.15 | 2 | 0.15 | 24 | 10 | **1.4E-10** | 0 | 1 |
| Ectopic fat traits | | | | | | | | | | | | | | | | | | | | | |
| VAT | 18,312 | 82 | 21 | **9.6E-16** | 1 | 0.87 | 25 | 1 | 0.47 | 0 | 1 | 28 | 2 | 0.15 | 4 | 4.9E-03 | 24 | 1 | 0.46 | 1 | 0.46 |
| SAT | 18,206 | 82 | 24 | **2.8E-19** | 0 | 1 | 25 | 7 | **2.0E-06** | 0 | 1 | 28 | 3 | 0.03 | 0 | 1 | 24 | 6 | **2.2E-05** | 0 | 1 |
| VAT/SAT | 18,205 | 82 | 4 | 0.15 | 6 | 0.02 | 25 | 0 | 1 | 2 | 0.13 | 28 | 3 | 0.03 | 11 | **3.5E-11** | 24 | 0 | 1 | 11 | **4.4E-12** |
| PAT | 11,616 | 82 | 8 | 1.1E-03 | 0 | 1 | 25 | 1 | 0.47 | 0 | 1 | 28 | 1 | 0.51 | 0 | 1 | 24 | 1 | 0.46 | 7 | **1.5E-06** |
| Cardio-metabolic traits and diseases | | | | | | | | | | | | | | | | | | | | | |
| CAD | ~185,000 | 81 | 15 | **1.6E-09** | 2 | 0.60 | 25 | 4 | 3.2E-03 | 3 | 0.02 | 28 | 2 | 0.15 | 5 | 5.9E-04 | 23 | 0 | 1 | 6 | **1.7E-05** |
| MI | ~170,000 | 81 | 9 | **2.0E-04** | 3 | 0.33 | 25 | 3 | 0.02 | 4 | 3.2E-03 | 28 | 0 | 1 | 2 | 0.15 | 23 | 0 | 1 | 5 | **2.3E-04** |
| HDL-C | 187,135 | 81 | 4 | 0.15 | 36 | **9.1E-36** | 25 | 2 | 0.13 | 6 | **2.9E-05** | 28 | 12 | **1.2E-12** | 2 | 0.15 | 24 | 15 | **9.8E-19** | 2 | 0.12 |
| LDL-C | 173,058 | 81 | 8 | 9.7E-04 | 4 | 0.15 | 25 | 2 | 0.13 | 2 | 0.13 | 28 | 1 | 0.51 | 9 | **1.7E-08** | 24 | 1 | 0.46 | 6 | **2.2E-05** |
| TG | 177,829 | 81 | 33 | **2.2E-31** | 2 | 0.60 | 25 | 4 | 3.2E-03 | 2 | 0.13 | 28 | 0 | 1 | 16 | **5.3E-19** | 24 | 0 | 1 | 15 | **9.8E-19** |
| T2D | 69,033 | 82 | 18 | **1.8E-12** | 1 | 0.87 | 25 | 3 | 0.02 | 2 | 0.13 | 28 | 0 | 1 | 4 | 4.9E-03 | 24 | 1 | 0.46 | 10 | **1.4E-10** |
| FG | 46,186 | 82 | 7 | 4.5E-03 | 2 | 0.61 | 25 | 1 | 0.47 | 2 | 0.13 | 28 | 0 | 1 | 1 | 0.51 | 24 | 0 | 1 | 2 | 0.12 |
| FI | 38,238 | 82 | 18 | **1.8E-12** | 1 | 0.87 | 25 | 3 | 0.02 | 1 | 0.47 | 28 | 0 | 1 | 3 | 0.03 | 24 | 0 | 1 | 9 | **3.5E-09** |

BMI: Body mass index; WHR: Waist-hip ratio; WC: Waist circumference; HIP: hip circumference; HT: Height; WT: Weight; VAT: Visceral adipose tissue volume; SAT: Subcutaneous adipose tissue volume; PAT: Pericardial adipose tissue volume; CAD: Coronary Artery Disease; MI: Myocardial Infarction; HDL-C: High-Density-Lipoprotein-Cholesterol; LDL-C: Low-Density-Lipoprotein-Cholesterol; TG: Triglycerides; T2D: Type 2 Diabetes; FG: Fasting Glucose; FI: Fasting Insulin
The table shows results from binomial tests that were conducted to test the variants of each class for enrichment of nominally significant increasing or decreasing effects on various traits including anthropometric traits, impedance measures, ectopic fat traits and cardio-metabolic traits, and disease. The table shows the number of variants tested ($n_{Tested}$), the number of variants with nominally significant trait-increasing or decreasing effects ($n$) and the respective binomial P value ($P_{Binomial}$). Bold binomial P values indicate significant enrichment ($P_{Binomial}$ <0.05/168, Bonferroni-corrected for 168 binomial tests) of nominally significant trait-increasing or trait-decreasing effects on the respective lookup trait

**A wrap-up of the class-specific adiposity phenotypes.** When summarizing the results of our data and analysis, we are able to characterize our four adiposity genetics classes with regard to anthropometry, fat depots, metabolic consequences, and implicated pathways (Supplementary Table 2): (i) BMI + WHR + alleles increased waist, hip, SAT, VAT as well as T2D and CAD risk consistent with the observed adverse lipids and insulin profile. This would be in line with a biological model of a CNS-triggered increase in fat mass and a metabolically unfavorable genetic pre-disposition to store fat subcutaneously and viscerally (metabolically unfavorable adiposity, e.g., MC4R and FTO[22,23]). (ii) BMIonly + alleles presented a similar pattern with increased hip, waist, and SAT, but without VAT storage consistent with an observed neutrality toward T2D or CAD (except TCF7L2, Fig. 4d, Fig. 5b). This would be in line with a CNS-triggered increase in fat mass and a metabolically neutral genetic pre-disposition to store fat subcutaneously rather than viscerally on both belly and lower body (metabolically neutral adiposity). (iii) WHRonly− alleles increased hip, but decreased waist, without any effect on BMI, total fat mass, VAT or SAT, but a decreased VAT/SAT ratio and a tendency toward a favorable metabolic profile (e.g., loci around PPARG, PLXND1, MAP3K1, RSPO3, PLXND1, JUND, Fig. 4d/e). This would be in line with a mechanism of fat redistribution as described for PPARG or RSPO3[24–26] (redistributing adiposity). At least one WHRonly− variant pointed to a different mechanism of enhanced bone growth: the variant near GDF5-UQCC is a known height locus[11] and got grasped by WHRonly− due to increased hip probably from bone growth rather than adiposity. For the genes within WHRonly− signals, we found enrichment of expression in digestive systems in DEPICT and FUMA analyses. (iv) BMI + WHR− alleles increased hip and SAT, but had no effect on waist or VAT, and a markedly favorable metabolic profile (metabolically rather favorable adiposity, e.g., GRB14-COBLL1). Our Mendelian Randomization approach[27] restricting the instruments to the BMI + WHR− variants showed that their BMI-increasing effect was causally linked to a favorable metabolic profile, particularly decreased risk of T2D and CAD. We also showed that the BMI increase of BMI + WHR− variants was causally linked to increased hip circumference and SAT, but had no effect on waist circumference or VAT. This would be in line with a direct beneficial effect of SAT stored on hip, possibly through adipokines[12], for this subtype of adiposity effects.

## Discussion

Our investigation demonstrated that a classification of genetic adiposity variants based on their co-association with BMI and WHR characterized distinct anthropometry and different modes of fat deposition. Importantly, our four classes help distinguish metabolically unfavorable (BMI + WHR +) and *metabolically rather favorable adiposity*(BMI + WHR−) at high precision that prompted the identification of 16 loci for favorable adiposity including 10 novel compared with previous work[24,28]. The focus on one of the four classes (WHRonly−) enabled us to reveal the digestive systems as a pathway for obesity genetics that extends upon previous work highlighting neural and adiposite/insulin pathways[5,6]. Our work has implications for adiposity research and GWAS methodology.

With regard to adiposity research, our work links to previous work that separated between BMI-scan identified and WHRadjBMI-scan identified adiposity variants and highlighted differential pathways, neural (BMI) versus adipose/insulin (WHRadjBMI)[5,6]. Further work used BMI-derived and WHRadjBMI-derived variants to demonstrate a causal

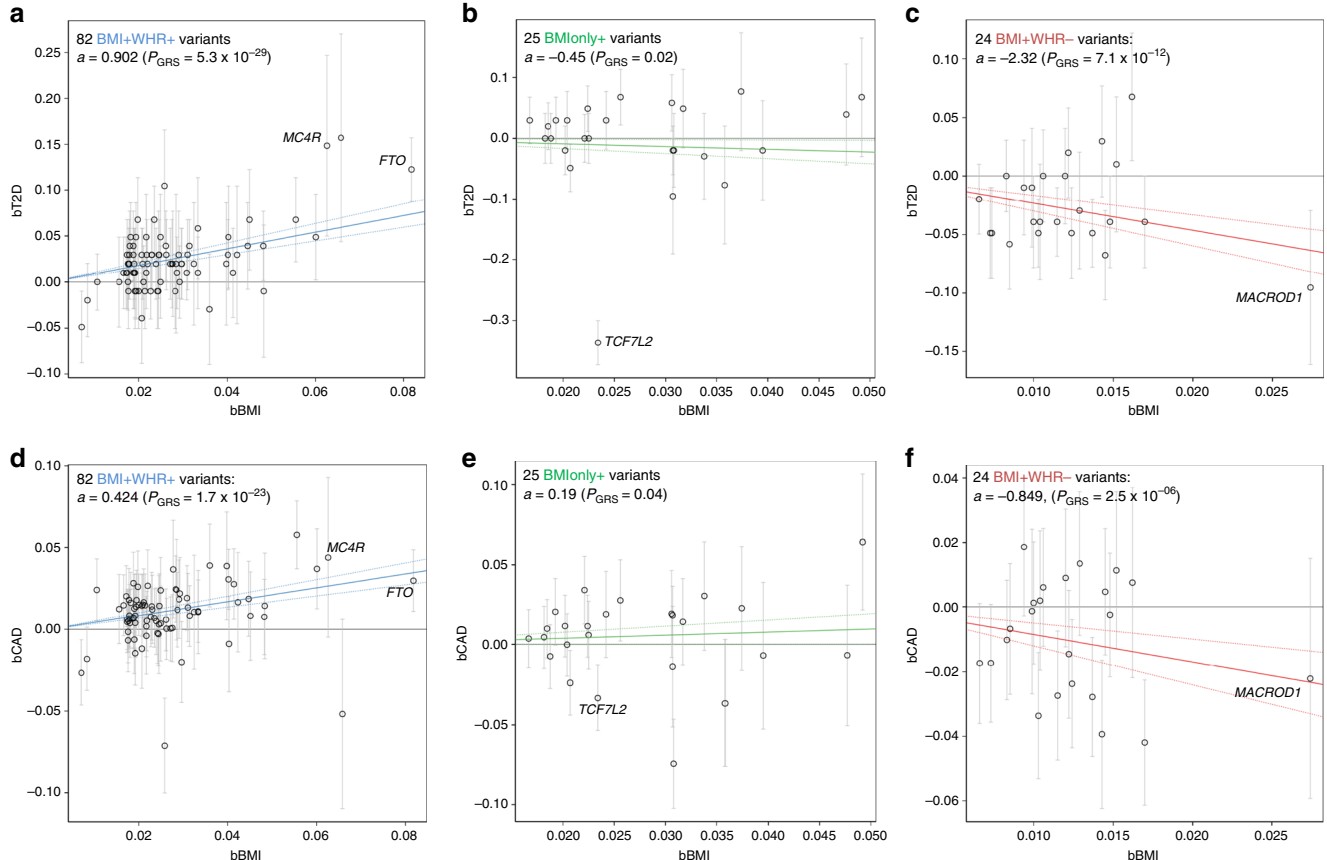

**Fig. 5** Different disease implications. Shown are the class-specific variants' co-associations for BMI and T2D or CAD (from DIAGRAM[15] and CARDIoGRAMplusC4D[18]) and the meta-regression line that can be interpreted as the association of the genetic risk score of BMI-increasing alleles (GRS) and disease (slope estimate indicated as "$a$", p value as $P_{GRS}$): **a** $BMI + WHR +$ variants on T2D, **b** $BMIonly +$ variants on T2D; **c** $BMI + WHR-$ variants on T2D, **d** $BMI + WHR +$ variants on CAD, **e** $BMIonly +$ variants on CAD; **f** $BMI + WHR-$ variants on CAD. Although the higher GRS for BMI is significantly associated with increased T2D and CAD risk for $BMI + WHR +$ variants, it is associated with decreased T2D and CAD risk for $BMI + WHR-$ variants

relationship of general or central obesity, respectively, with T2D risk via Mendelian Randomization[8,22,29]. In our approach, we define four classes of adiposity genetics based on the variant's co-association with BMI and WHR that generally, with some exceptions, split BMI-derived variants into two groups ($BMI + WHR +$, $BMIonly +$) and WHRadjBMI-derived variants into two groups ($BMI + WHR-$, $WHRonly-$). Our classification based on nominal significance of BMI and/or WHR association is straightforward and easy to apply. Certainly, there are other methodological approaches for clustering or evaluating multivariate effects worthwhile to be explored in the future[30].

Investigating these four classes separately, we find the following aspects of adiposity mechanisms: (i) Our classes $BMI + WHR +$ and $BMIonly +$ differentiate BMI-derived effects between those that involve VAT and increased disease risk from those that do not (with few exceptions). (ii) Our class $BMI + WHR-$ distinguishes effects of pure hip increase (without any effect on waist) from those with altered waist ($BMI + WHR +$, $BMIonly +$, $WHRonly-$). The BMI-increasing alleles of $BMI + WHR-$ variants are all hip-increasing and jointly show a substantial reduction of disease risk (T2D OR = 0.10, CAD OR = 0.43). This finding contributes to the considerable debate on whether the acknowledged importance of SAT stored on the lower body[31] stems from its role as a reservoir to avoid fat storage on more detrimental places like VAT[32] or from a directly beneficial effect from lower body fat itself[33], as the $BMI + WHR-$ variants show no effect on waist circumference, their metabolically beneficial effect can only stem from a directly favorable effect from hip

increase, but not from a less-detrimental storage compared with central body fat (then the other allele would be waist-increasing). This also emphasizes the role of WHRadjBMI as a relative measure of lower body fat that is missed when focusing on WHRadjBMI as a measure of central obesity. (iii) For the $WHRonly-$ class, which focuses on a subset of WHRadjBMI variants, significant expression enrichment identifies digestive systems as a pathway for obesity genetics via two independent methods utilizing two independent data sets (FUMA and DEPICT). The fact that this enrichment emerged only when restricting to $WHRonly-$ loci, but not when analysing all WHRadjBMI loci together, suggests that $WHRonly-$ loci capture an adiposity subtype that is diluted in the larger set of WHRadjBMI loci. Still, further data and experiments will be necessary to determine the mechanisms through which these variants can be linked to transcriptional regulation in digestive systems. Altogether, we conclude that our four classes capture distinct anthropometry and fat depots and help distinguish important adiposity mechanisms (Fig. 7) that were missed by the previous separation into only two groups. Although there are exceptions within classes and metabolic implications have to be validated by locus, this differentiation can help prioritize adiposity loci for therapy development pipelines[34].

There have been different approaches to capture favorable adiposity. Among the 11[24] or 53[28] loci previously identified for insulin resistance and put into context with favorable adiposity, 7 or 13 loci, respectively, capture favorable adiposity effects in the here utilized data following a definition where the BMI-increasing

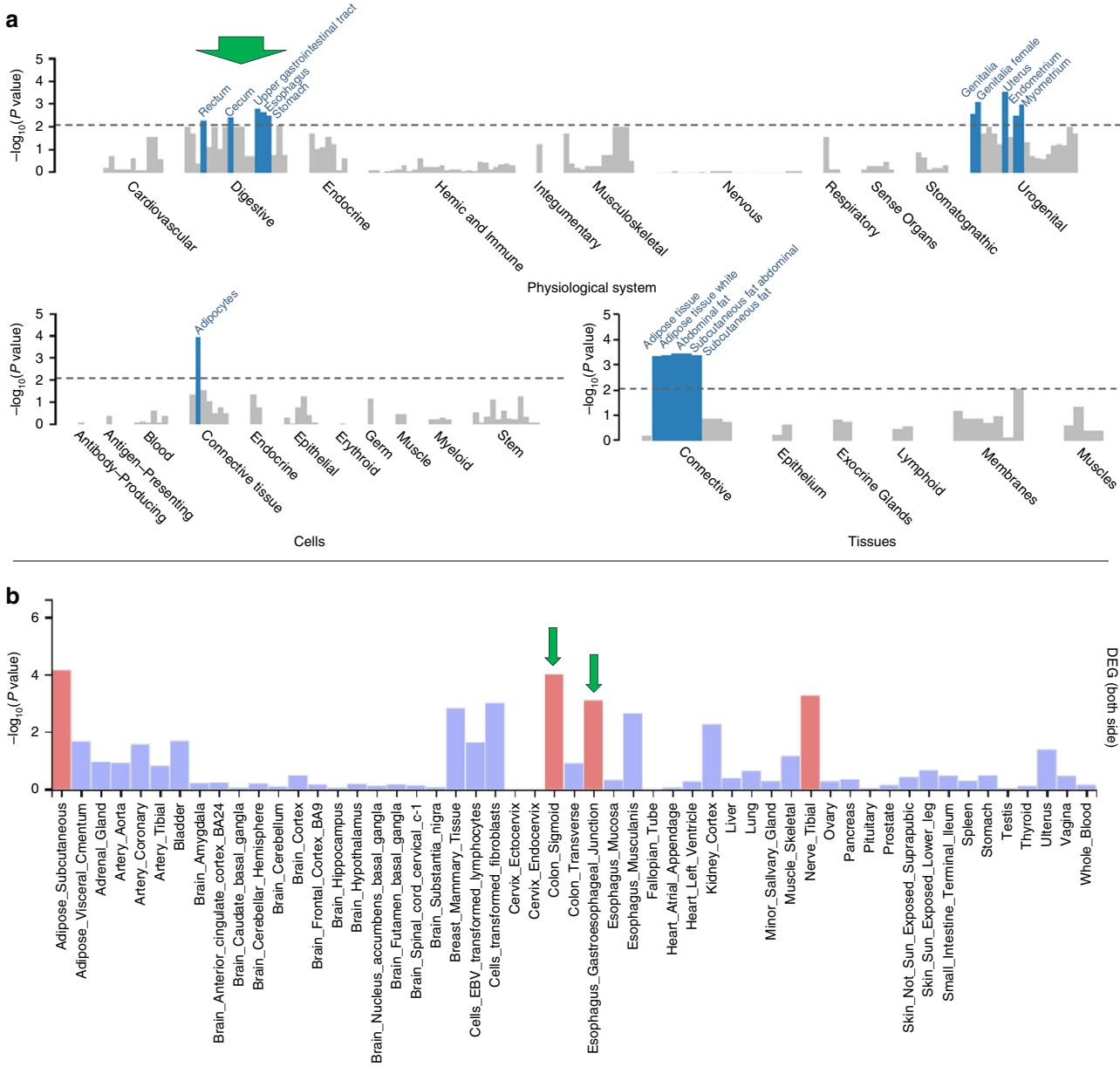

**Fig. 6** Tissue-specific gene expression for *WHRonly−* variants. Shown are results of DEPICT and FUMA tissue-specificity analyses based on variants that were selected from GWAS-only meta-analyses of GIANT ($P < 10^{-5}$) and that were classified as *WHRonly−*. Significant results within the digestive system are marked with green arrows. **a** DEPICT results for *WHRonly−* with significant enrichments highlighted in blue (FDR < 5%). Results are grouped by type and ordered alphabetically by MeSH term within a specific system, cell type, or tissue (details in Supplementary Data 9). Results for the other three classes showed no significance with DEPICT (Supplementary Figure 6). **b** FUMA results with significant enrichments highlighted in red (adjusted *P* < 0.05, Bonferroni-corrected, details in Supplementary Data 13). The -log10(*P*values) in the graph refer to the probability of the hypergeomteric test. Results for the other three classes showed only little enrichment with FUMA (Supplementary Figure 8)

allele ($P_{BMI} < 0.05$) shows decreased risk of T2D or CAD ($P_{T2D}$ or $P_{CAD} < 0.05$, no increased risk in either). To be comparable, we derived 1 Mb regions around our 159 lead variants resulting in 117 distinct regions. Of these 117 regions, excluding *TCF7L2* owing to its extreme T2D risk (and potential index event bias in the BMI-association[5,35]), 16 regions contained one of our 159 signal variants with a favorable adiposity effect. Of these 16 regions, 10 were novel compared with previous work[24,28], 10 were classified as *BMI + WHR−* including seven novel (Supplementary Data 14). We were thus able to increase the number of loci for favorable adiposity by 50%.

With regard to GWAS methodology, we confirm several points brought up by Aschard and colleagues[1]: (1) WHRadjBMI effects

differ from WHR effects by $−r*b_{BMI}$ (with *r* being the phenotypic correlation). We provided empirical data that WHR effects can be effectively computed from WHRadjBMI and BMI effects in a single study and in a meta-analysis setting (Fig. 3). (2) WHRadjBMI-derived genetic effects are not necessarily "WHR effects independent of BMI", as WHRadjBMI-derived variants can have effects on BMI as shown in theory (see (1)) and observed in our data (*BMI + WHR−*, some in *BMI + WHR +* ). (3) WHRadjBMI GWAS enrich for WHR effects with simultaneous effects on BMI into the opposite direction, which are exactly the effects in the *BMI + WHR−* class.

However, there might have been some misconceptions about the implications of these points that we believe our results and

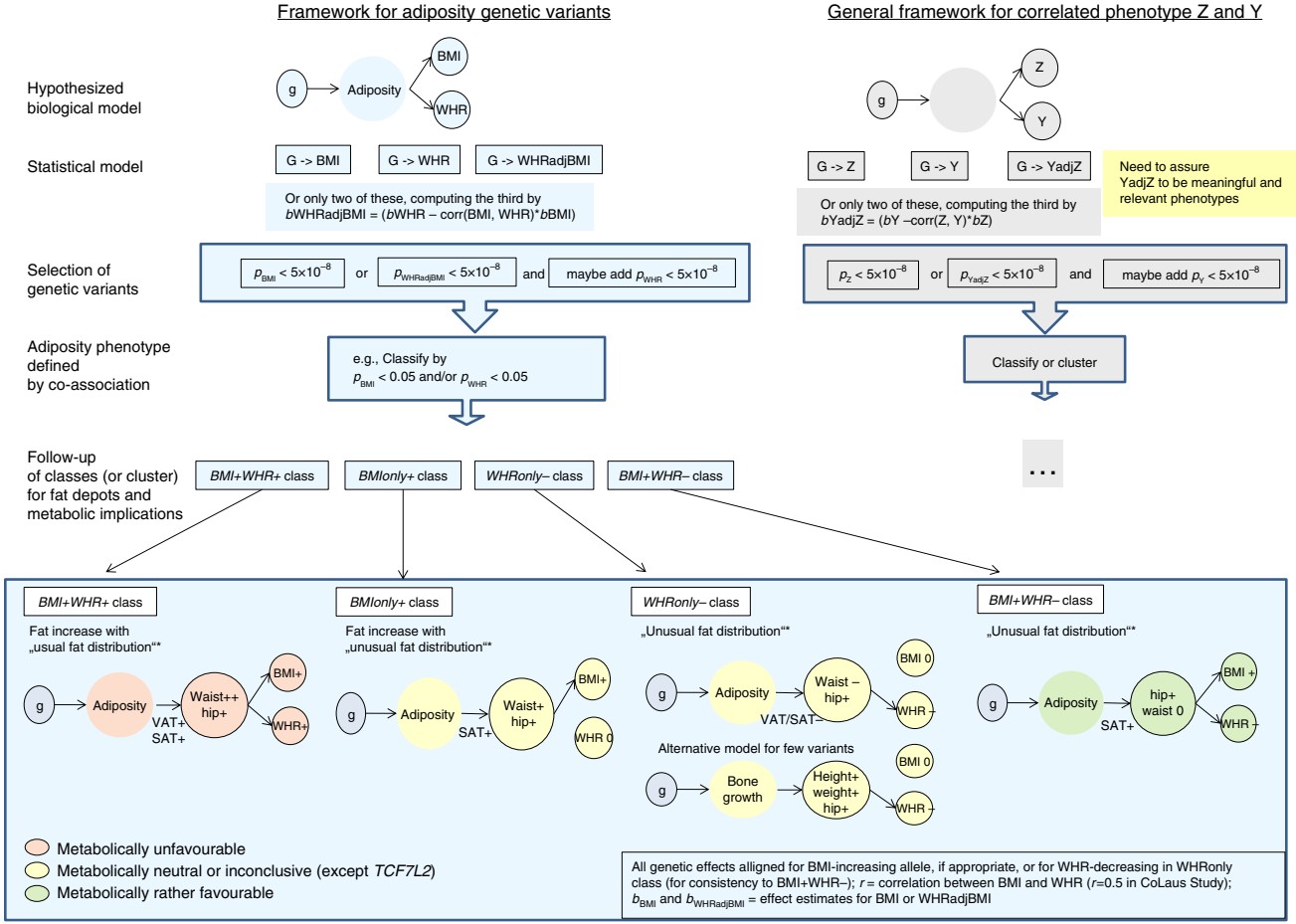

**Fig. 7** Approach, biological models, and general framework. The figure depicts our joint view on adiposity traits, the resulting hypothesized biological models for adiposity genetics, and a potential general framework

approach can help resolve: (a) whereas WHRadjBMI effects are not "independent of BMI", our WHRadjBMI-derived variants are distinct from BMI-derived variants with regard to their position on the $b_{WHR}-b_{BMI}$ plane (Fig. 2a). With increasing sample sizes, WHRadjBMI− and BMI-derived variants cannot be expected to keep this distinction and several variants will be captured by both screens, which will result in overlapping biology. However, this can be resolved by adopting our joint view and classifying the variants by their position on the $b_{WHR}-b_{BMI}$ plane. (b) WHRadjBMI-derived variants were suspected to yield spurious association. When considering spurious association of WHRadjBMI-derived variants in the sense that such variants end up having no effect on WHR but only on BMI, we found no such alleged spurious association: all of our WHRadjBMI-derived variants showed an effect on WHR in GIANT data (at nominal significance) and in UKBB data at Bonferroni-corrected significance (UKBB $N = 336,107$, $P_{WHR} < 0.05/53$, Supplementary Data 2). Even if there were WHRadjBMI-derived variants with an effect only on BMI (and such variants will be detected eventually when GWAS sample size increases), such variants would be part of "adiposity genetics" in a joint view of BMI- and WHRadjBMI-derived variants. (c) When considering spurious association of WHRadjBMI variants in the sense that such variants end up having no effect, neither on WHR nor on BMI, this would be a real concern as these would be variants without adiposity effect. As indicated above (see (b)), there is no such spurious association in the data (all our WHRadjBMI-derived variants have an effect on WHR in GIANT and UKBB) and, in theory (Supplementary Note 4). (d) The "bias-correction" given by Aschard and

colleagues is simply an estimation of the WHR effect from WHRadjBMI and BMI effect estimates (see (1) above). However, the term "bias-correction" is misleading as the WHRadjBMI effect sizes are not a nuisance (as effects with bias usually are), but capture the extent to which the observed WHR effect differs from the expected given the variant's effect on BMI. This is, in the context of adiposity, a relevant quantity as an unexpected change in fat distribution given the change in BMI can mark metabolically relevant conditions (as an extreme, e.g., lipodystrophy) – WHRadjBMI is thus a meaningful phenotype.

We believe that, possibly due to the misinterpretation of the work of Aschard and colleagues, WHRadjBMI GWAS was perceived as treacherous and less useful than a WHR GWAS. However, we have shown that omitting the WHR-scan would have missed only four variants, whereas an omission of the WHRadjBMI-scan—out of a fear of bias and spurious association—would have missed 27 adiposity genetics signals. We have also shown that WHR-derived variants lack any distinct pathway pattern, whereas WHRadjBMI-derived variants are health-relevant (some confer favorable adiposity, some fat redistribution) and pick up important biology (expression in adipose/insulin and digestive systems). We conclude that, in this example of adiposity genetics, the adjusted-model trait GWAS has advantages over the unadjusted trait GWAS. This does not mean that each adjusted-model trait GWAS is useful; this has to be evaluated on a case-by-case basis.

Our recommendations for future GWAS on adiposity genetics are as follows: (A) if only two GWAS scans (rather than three) are feasible, stick with the BMI- and the WHRadjBMI GWAS, (B)

WHR effects can be computed by WHRadjBMI and BMI effects, if not available otherwise, (C) a joint view of BMI- and WHRadjBMI-derived variants on the $b_{BMI} - b_{WHR}$ plane provides a clearer view on the underlying anthropometry than separating between BMI- and WHRadjBMI-derived genetics, and (D) a classification using the variants' position on the $b_{BMI} - b_{WHR}$ plane can serve to differentiate adiposity mechanisms and metabolic health. Particularly, $BMI+$ $WHR-$ variants can be used effectively to search for favorable adiposity effects. Our approach of joining WHRadjBMI and BMI-derived variants (with or without adding WHR-derived variants), rather than disentangling[5,6,22,23], and linking the variants' positions on the $b_{BMI} - b_{WHR}$ plane to metabolic implications (Fig. 7) helped resolve some of the appreciable uncertainty about the utility of WHRadjBMI variants' effects. Our approach can be generalized to a framework for other settings, where two related phenotypes Y and Z are two correlated measures of a latent heritable entity (here: adiposity) and where the adjusted-model trait YadjZ is a meaningful phenotype. An important lesson learned is to view adjusted-model traits and the co-association of related phenotypes as a powerful tool to identify important biology, but to interpret them with great care keeping in mind the underlying biological models.

In summary, our approach and results provide insights into adiposity subtypes and an example for a co-analysis of related phenotypes including adjusted-model traits to help reveal new biology.

## Methods

**The GIANT consortium data**. Our evaluation was based on genome-wide association meta-analysis results for BMI, WHR and WHRadjBMI (from 2015) that are publically available from the GIANT consortium website (www.broadinstitute.org/collaboration/giant)[5,6]. We used sex-combined, European ancestry meta-analysis results including up to 322,154 persons (114 studies), 212,248 (101 studies), or 210,088 (101 studies) for BMI, WHR, or WHRadjBMI, respectively. In brief, in each study, inverse-normal transformed residuals were calculated from regressing BMI and WHR on age, age[2], and other study covariates like principal components—and on BMI to derive WHRadjBMI. The genetic effect estimates on BMI, WHR, and WHRadjBMI were obtained from inverse-variance weighted meta-analyses of study-specific genetic effect estimates on BMI, WHR, or WHRadjBMI, respectively. Genome-wide association studies were either based on Hapmap-imputed SNPs (~2.8 M SNPs) or on genotyped Metabochip SNPs (~190 K SNPs). For further details, see the study descriptive in published literature for BMI[5], WHR, and WHRadjBMI[6]. Informed consent was obtained from all study participants, and study protocols were approved by the local ethics committees.

**Integrative genome screen on BMI, WHR, and WHRadjBMI**. We excluded SNPs with < 10,000 individuals contributing to the respective meta-analysis and SNPs on sex chromosomes. For each of the three traits, we first selected all variants at genome-wide significance ($P < 5 \times 10^{-8}$). We then combined these three sets of SNPs derived across the three traits yielding a set of 2589 SNPs. We clumped these into 159 non-overlapping, independent regions using a combined distance- and linkage disequilibrium-based criterion (< 500kB to either side; and $r^2 > 0.1$). The lead SNP of a region was defined as the SNP with the smallest association $P$-value within the region (no matter from which trait the SNP was derived). The "traits of a region" were defined as the union of traits contributing with genome-wide significance across the SNPs of the respective region. We used PLINK[36] and Easy-Strata[37] for the clumping and for the comparison across traits. Of note, our clumping strategy will yield a different number of independent signals compared with the two GIANT studies that applied a distance-only criterion (no r² threshold, but consecutive conditional analyses) and used multiple ancestries as well as sex-specific results to select significantly associated variants.

**Adiposity traits from the UK Biobank**. We used data from the final release of the UK Biobank ($N$ up to 336,107) to follow-up our WHRadjBMI-derived ($P_{WHRadjBMI} < 5 \times 10^{-8}$ in GIANT) for their association with BMI and WHR. The results for WHR and BMI were downloaded from the GeneAtlas[9] website (http://geneatlas.roslin.ed.ac.uk/downloads/?traits = 92) and from the Neale Lab website (https://sites.google.com/broadinstitute.org/ukbbgwasresults/), respectively.

For the comparison of estimated and re-computed WHR effects, we used data from the UK biobank interim release ($N$ up to 116,295). For this, we utilized linear regression to obtain residuals of BMI or WHR adjusted for age, age[2] (for WHR: additional for BMI), five ancestry principal components, and batch indicators, and

used these residuals to derive genetic association (as in GIANT). This was done by sex and results meta-analyzed.

**Anthropometric trait lookups from the GIANT consortium**. To investigate the identified loci for their effects on waist circumference, hip circumference (HIP), weight (WT), and height (HT), we utilized the publically available meta-analysis summary statistics from the GIANT consortium (www.broadinstitute.org/collaboration/giant): From the 2015 round of meta-analyses, the sex-combined results for WC[6] (up to $N = 232,083$), HIP[6] (up to $N = 213,028$) and HT[38] (up to $N = 253,239$) as well as sex-specific results for BMI[5] and WHR[6]; and from the 2013 round of sex-specific meta-analyses, the sex-specific results for WT[8]. In order to derive sex-combined meta-analysis results for WT (up to $N = 125,943$), we conducted a fixed-effect inverse-variance weighted meta-analysis of the two sex-specific WT results. Again, the GIANT consortium results were based on inverse-normal transformed (or standardized for HT) phenotypes in the study-specific analyses.

**Fat compartments using impedance measures from UK Biobank**. We investigated the effects of the identified variants on impedance-derived measures of body, leg, and trunk fat mass using data from the UK Biobank. We analyzed up to 114,367 unrelated samples of genetically determined European ancestry. Linear regression analyses were applied for the 157 available SNPs including the covariates age, sex, age[2], five ancestry principal components, and batch indicators. Sex-stratified analyses were conducted and subsequently meta-analyzed.

**Fat compartments using imaging from ectopic fat consortium**. We evaluated the effects of the identified variants on computed tomography and magnetic resonance imaging derived measures of ectopic fat volumes as described previously[4]. This data from the Ectopic fat consortium is publically available (www.nhlbi.nih.gov/research/intramural/researchers/ckdgen). We utilized the meta-analysis summary statistic for (SAT, up to $N = 18,206$), visceral adipose tissue (VAT, up to $N = 18,312$) and pericardial adipose tissue (PAT, up to $N = 11,616$) as well as for the ratio of VAT and SAT (VAT/SAT, up to $N = 18,205$). For comparison reasons, we computed beta-estimates (assuming a standardized outcome) from the publically available Z scores using beta $= Z/\sqrt{(N * 2* eaf * (1-eaf))}$, where eaf is the allele frequency of the effect allele, $N$ is the sample size of the meta-analysis and $Z$ the corresponding (and provided) Z score.

**Cardio-metabolic traits and diseases lookups**. To investigate the identified variants for their effects on other metabolic traits, we utilized publically available meta-analysis summary statistic from several genomic consortia: T2D (up to $N = 69,033$) from DIAGRAM[15] (www.diagram-consortium.org/downloads.html), fasting insulin (FI, up to $N = 38,238$) and fasting glucose (FG, up to $N = 46,186$) from MAGIC[17] (www.magicinvestigators.org/downloads/), for triglycerides (TG, up to $N = 177,829$), HDL-C (up to $N = 187,135$) and LDL-cholesterol (up to $N = 173,058$) from Global Lipids Genetics Consortium[16] (csg.sph.umich.edu/abecasis/public/lipids2013/), and for myocardial infarctions (MI, up to $N~170,000$) and CAD (up to $N~185,000$) from the CARDIoGRAMplusC4D[18] (www.cardiogramplusc4d.org/data-downloads/). Again, if not available from the downloaded data, we computed beta-estimates (assuming a standardized outcome) from the publically available z scores according to the formula provided before.

**Enrichment analyses**. We applied binomial tests to evaluate whether the variants in each class were enriched for nominal significant effects on anthropometric, impedance, ectopic fat, or cardio-metabolic traits. We applied a conservative Bonferroni-corrected significance level to the binomial tests ($P_{Binomial} < 0.05/168$, corrected for four classes × 21 traits × 2 direction of effects).

**Mendelian randomization analysis by class**. We also conducted class-specific inverse-variance weighted summary statistic based Mendelian randomization analysis[27], in order to explore the causal implication of BMI increase on 20 traits[27,29,39] (including anthropometric, fat depot, and metabolic traits) separately by each of the adiposity subtypes. This way we estimated the causal effect of the BMI increase on all 20 traits by restricting the instruments (i.e., the genetic variants) to one class. We excluded the WHRonly− variants, which are no effective instruments for BMI. We applied a conservative Bonferroni-corrected significance level to the meta-regression results ($P_{GRS} < 0.05/60$, corrected for three classes MR 20 traits).

**DEPICT analyses**. In order to search for enriched pathways among the genes beneath association signals, there are several tools available, with little evidence, which one is superior. We utilized DEPICT[19] to test whether genes harboring associated variants were enriched for genes with expression effects in different tissue, cell type and physiological system. More specifically, DEPICT tests whether the genes in associated regions are highly expressed in any of the 209 MeSH annotations for 37,427 microarrays on the Affymetrix U133 Plus 2.0 Array platform. DEPICT version 1 rel194 was downloaded from https://data.broadinstitute.org and updated using scripts from GitHub (https://github.com/perslab/depict). To

prevent bias in the enrichment analysis results owing to the customized design of the Metabochip, we restricted the DEPICT analyses to variants with pronounced association in the GWAS-only meta-analyses (i.e., excluding Metabochip data) as done previously[5,6]. These data were available to us through our collaboration with GIANT. Our analysis was based on all variants reaching $P < 1 \times 10^{-5}$ in at least one of the GWAS-only meta-analyses for BMI, WHR, or WHRadjBMI. We clumped the GWAS-only variants based on linkage disequilibrium of $r^2 > 0.1$ from 1000 Genomes[40] data and 500 kb flanking regions to obtain lists of independent SNPs. We conducted DEPICT analyses separately for the loci that were identified with the BMI-scan, the WHR-scan, or the WHRadjBMI-scan (scan-specific pathway analyses). We also conducted the enrichment analysis separately for the variants in each of the four classes $BMI + WHR +$, BMI only, WHR only, and $BMI + WHR-$ (class-specific pathway analyses) using all at $P < 10^{-5}$ (GWAS-only) and then classifying variants based on the combined GWAS and Metabochip meta-analysis results. For each DEPICT analysis, the list of utilized SNPs were merged with overlapping genes utilizing precomputed gene regions based on 1000 Genomes project variants. SNPs within the major histocompatibility complex region on chromosome 6, base pairs 25,000,000–35,000,000, were excluded. DEPICT analyses were conducted using the following parameters: 100 repetitions to compute FDR and 1000 permutations based on 500 null GWAS to compute $P$ values adjusted for bias due to gene length. A total of 10,968 reconstituted gene sets were used for the enrichment analysis. Tissue/cell type and Physiological system enrichment plots were generated using R-scripts on the basis of the R-script provided with DEPICT. Sensitivity analyses for the $WHRonly$- class were conducted that (1) excluded all known height-associated loci from (Wood et al. 2014), (2) excluded variants harboring the $RSPO3$ gene and (3) utilized a wider locus definition criterion that was based on distance-only ( ± 500 kb).

**FUMA analyses**. We utilized the FUMA[20] web tool (http://fuma.ctglab.nl/) to investigate whether gene sets harboring our class-specific association signals were enriched for genes with expression signals in specific tissues of the GTEx v6 data[21]. FUMA applies a two-sided $t$-test to infer whether gene sets are differently expressed (up- or downregulated) in any tissue compared to all others tissues. Differentially expressed gene (DEG) sets are utilized by FUMA that were pre-calculated based on GTEx v6 expression values. DEGs in specific tissues compared with others reflect those gene sets with Bonferroni-corrected enrichment $P$ values ≤ 0.05 and absolute log fold change ≥ 0.58. To prevent from bias through the customized design of the Metabochip, we selected associated variants ($P < 10^{-5}$ for BMI, WHR, or WHRadjBMI) from GWAS-only meta-analyses (excluding Metabochip studies) of the GIANT consortium. We then utilized GWAS + Metabochip meta-analyses results for the classification, which comprise a larger sample size and thus higher accuracy for the classification. The applied input data sets of association results were identical to the ones that were applied to the DEPICT tissue-specificity analysis. For each class, genes harboring the respective associated regions were tested by FUMA against each of the DEG sets using the hypergeometric test, whereas background genes were those with an average Reads per kilo base per million mapped reads > 1 in at least one of the 53 GTEx tissues and exist in the list of background genes. We included all gene types in the prioritization that were available through FUMA and used default FUMA input values, e.g., $r^2 < 0.6$ to define locus regions around the associated regions.

**Data availability**. Summary genetic association results that are used as basis for this study are available on the GIANT consortium website (http://portals.broadinstitute.org/collaboration/giant/) for BMI ('SNP_gwas_mc_merge_nogc.tbl.uniq.gz')[5], WHR ('GIANT_2015_WHR_COMBINED_EUR.txt.gz')[6], and WHRadjBMI ('GIANT_2015_WHRadjBMI_COMBINED_EUR.txt.gz')[6]. All other data that support the findings of this study are available from the corresponding author upon reasonable request. Our integrative analysis of publically available GIANT data was performed with the open-source R package EasyStrata[37]. EasyStrata-scripts are available for download from www.genepi-regensburg.de/easystrata.

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

## Acknowledgements

This research has been conducted using the UK Biobank Resource. I.M.H. received funding from the Bundesministerium für Bildung und Forschung (BMBF, 01ER1206, 01ER1507) and from the National Institutes of Health (NIH, R01DK075787). Z.K. received financial support from the Swiss National Science Foundation (31003A_169929) and SystemsX.ch (51RTP0_151019). R.J.F.L. received support from the National Institutes of Health (NIH, R01DK107786, R01DK110113, U01HG007417). This work was supported by the German Research Foundation (DFG) within the funding programme Open Access Publishing.

## Author Contributions

T.W.W., F.G., S.H., M.E.Z., R.J.F.L., Z.K., and I.M.H. wrote the manuscript; T.W.W., Z.K., and I.M.H. conceived and designed the project; T.W.W. and Z.K. conducted association analyses; T.W.W., S.H., and M.E.Z conducted DEPICT analyses.

## Additional information

**Competing interests:** The authors declare no competing interests.

