## [Peer Review File · Nature Communications]

Reviewer #2 (Remarks to the Author):

The authors have done a good job in addressing my previous comments. However, I have several remaining concerns.

Abstract. I found the revised abstract difficult to follow. The first sentence didn't make sense to me, and I found it very separate from the remainder of the abstract. The use of the four categories is not helpful without explanation as to what the specific categories are.

Line 82. It is not clear what CoLau is – better to give r^2 across studies (either as a point estimate or range)?

Line 115. The authors state that all WHRadjBMI derived variants have an effect on WHR – at what level of significance, and is the effect always stronger than that on BMI?

Line 120. For WHRadjBMI, several loci have highly differentiated effects between males and females. Do female-specific loci always fall into the same category, and if they do, is the same true for the few male-specific loci?

Line 170. I found this paragraph very difficult to follow. I think it would be worth adding a couple of sentences about the previous study that attempted to identify/classify favourable adiposity loci. Maybe this would be better removed from the results, and just left in the discussion.

Line 184. Excluding MetaboChip data – I was confused by this, and it was not clear how the data used for these analyses differed from the original GIANT studies used for classification (and why this was necessary).

Line 199. I didn't understand what these different exclusions meant, and why they were being performed.

Line 202. For the eQTL analyses, why use $D\text{-prime}=1$ as a threshold. The more relevant metric is r^2 . Furthermore, the fact that a variant is also an eQTL does not mean that the trait association signal and eQTL signal are coincidental.

Line 212. The "wrap-up" paragraph would be better in the discussion.

Line 272. I didn't understand where the various numbers of loci were coming from – please clarify.

Line 302. There is a point (c), but I couldn't see point (b)!

Reviewer #3 (Remarks to the Author):

The revised manuscript from Winkler and colleagues satisfactorily addresses some but not all reviewer comments. Specifically, the report better addresses one concern of Aschard et al, but the conclusion that this analysis identifies novel biology is still weak and not well supported by the data. Some changes to the manuscript raise new concerns.

1. The new analyses of UK Biobank data to address collider bias provide a valuable contribution to the literature, especially showing that analysis of WHRadjBMI did not generate 'spurious' associations with the adjusted trait, even with improved power. However, the text could still be improved to better describe this contribution.

- Introduction lines 57-58 are need to more clearly explain the concern being addressed here
- The new Discussion on pages 9-10 is quite difficult to follow and needs clarification. For example, lines 294-295, how do the data confirm that WHRadjBMI GWAS may include variants with only a BMI effect when no such examples are found? Separate the supporting results from opinion. Also, line 296 has point (a) and line 302 has point (c), but there is no point (b). Consider further paragraph divisions to separate points, and further attention to clear sentence grammar and paragraph structure.

2. Interpreting loci based on the direction of effect on both BMI and WHR, dividing each group in two, seems like it could be valuable. However, the additional analyses regarding digestive and urogenital systems, providing lists of gene names and some eQTL lookups without further interpretation, did little to support the conclusion that these systems play a role in obesity genetics or that this analysis detects novel biology. The lack of substantial new biological insights remains a limitation of the work.

3. The description of WHRadjBMI as a measure of "unusual fat distribution" seems forced and too specific to extreme values. Analyzing any quantitative trait is not limited to revealing effects leading to unusual values but identifies effects leading to variation across the trait range. Here, usual is defined based trait correlation in all members of a population sample and unusual is based on an arbitrary p-value threshold. The analyses CAN DETECT unusual fat distribution, but the trait itself is not a measure of unusual fat distribution. The wording should be changed.

Minor point: the loci listed on line 107 with reference to Figure 1 are slightly different names than presented in the Figure: ANKRD65 vs ANKRD55 and CACLRL vs CALCRL

Reviewer #4 (Remarks to the Author):

Winkler et al use previously published GWAS meta-analysis results to explore the effect of BMI on waist-hip ratio signals, to partially rebut the critiques on employing "adjusted model" traits. They then use these comparisons to categorize BMI and WHR loci based on their co-association (or lack thereof) with the two traits to identify distinct subsets among, and then to characterize the biological relationships among and between these sets of loci. They argue that this differentiation helps identify novel physiological systems not previously implicated in obesity biology, namely digestive and urogenital systems.

This analysis is an interesting approach to both understanding the complexities of "adjusted model" traits, and for teasing apart the complex biology of obesity. While the authors generate little data on their own, their approach in bringing these data together to both address concerns about statistical methodology, and for understanding the results are valuable, and helpful in understanding both aspects. While their conclusions on the lack of biased and thus spurious findings for WHRadjBMI can only be as convincing as the current sample data allow, the consistency of the GIANT results with UKBB are reassuring. The robustness of their DEPICT and expression results are less convincing, and could be strengthened to convince the reader that these effects are specific to the subtype and thus real.

Comments:

1) It was imperative that the authors make a more explicit comment on the Aschard paper, and they make important points on the distinctions between how the Aschard findings have been interpreted and how these adjusted models function in reality. In particular, clarifying the past

nomenclature of "independent from BMI" is important. And the additional mathematical justifications, though mostly in the supplement, are important justifications for the intuitive, but common misconceptions. It is understandable, but too bad that the comparisons of expected to calculated results could not be highlighted more to more solidly justify the rest of the results.

2) The authors have clearly taken pains to make the classification of each locus into their 4 categories carefully. However, it is still extremely difficult to follow in the text, and distributions in Figures 1 A and C don't help to clarify much.

- How are none of the BMI+WHR- loci associated with BMI? in the Venn Diagram
- Why do the numbers in parentheses in 1C not add up to the expected totals?

3) I wonder if the choice of terminology is entirely appropriate. Why is one or another fat distribution "usual" or "unusual"? Is larger hip circumference (hip fat depots) really "favorable" or is it just relatively less bad? I worry that this may be somewhat misleading. Associations with shifts in SAT vs VAT are compelling, though.

4) A few comments on DEPICT analyses:

- I don't think the results in Supplement Fig. 6 are as convincing of previous findings as stated. It appears no findings are significant after subdividing SNPs into groups.
- Is there a way to use the set of 159 loci to test the robustness of the new tissue/system findings, to help convince that they aren't an artifact of dividing up the data arbitrarily? Could you permute sets the same size as the WHRonly- group to show that discovering new systems/tissues/cells is specific and not just a function of arbitrary subsetting and thus a chance finding?

5) Additionally, the GTEx result trying to support the digestive system is weak, and shows very little specificity for the set of variants in which those tissues were identified.

Point-by-point response to Reviewers

#####

Reviewer #2 (Remarks to the Author):

The authors have done a good job in addressing my previous comments. However, I have several remaining concerns.

+ Abstract. I found the revised abstract difficult to follow. The first sentence didn't make sense to me, and I found it very separate from the remainder of the abstract. The use of the four categories is not helpful without explanation as to what the specific categories are.

We thank the reviewer for the suggestions on the abstract. We have reworded most of the first sentence and further modified the abstract to make it clearer (line 26):

The genetics of related phenotypes is often tackled by analysing adjusted model traits, but such traits warrant cautious interpretation. Here, we adopt a joint view applied to adiposity traits. Analysing GIANT consortium data (~322,154 subjects), we classify 159 signals associated with any of body-mass-index (BMI), waist-to-hip-ratio (WHR), or WHR adjusted for BMI (WHRadjBMI) at $P < 5 \times 10^{-8}$, into four classes according to the direction of their effects on BMI and WHR. We show that our classes help differentiate adiposity genetics with respect to anthropometry, fat depots, and metabolic health. Class-specific Mendelian randomization reveals that variants with simultaneous WHR-decrease and BMI-increase are linked to metabolically favorable adiposity through beneficial hip fat. Class-specific enrichment analyses implicate digestive systems as novel pathway in adiposity genetics. Our results demonstrate that WHRadjBMI variants capture relevant effects of “unexpected fat distribution given the BMI” and that a joint view of related phenotypes genetics can help unravel important biology.

+ Line 82. It is not clear what CoLaus is – better to give r^2 across studies (either as a point estimate or range)?

We now mention that CoLaus is a population based study and have added a reference (line 98):

i.e. as expected by the phenotypic correlation r , e.g. $r=0.5$ in the population-based CoLaus study⁷

+ Line 115. The authors state that all WHRadjBMI derived variants have an effect on WHR – at what level of significance, and is the effect always stronger than that on BMI?

Indeed, the level of significance was missing at this point. The level of significance was 5%. The weakest WHR association for the 53 WHRadjBMI-derived variants was $P_{\text{WHR}} = 7.5 \times 10^{-3}$ (in GIANT data). We have clarified this in the manuscript (line 139):

All 53 WHRadjBMI-derived variants had nominally significant effects on WHR (i.e. no spurious associations, weakest WHR association observed in GIANT $P_{\text{WHR}}=7.5 \times 10^{-3}$, **Supplementary Table 1**).

Based on this comment and comments by other reviewers, we now also include association results for the 53 WHRadjBMI-derived variants for BMI and WHR from the final UK Biobank release

(N=336,107, new **Supplementary Table 2**). All of the 53 WHRadjBMI-derived variants ($P_{\text{WHRadjBMI}} < 5 \times 10^{-8}$ in GIANT) are significantly associated with WHR in the UK Biobank data not only at a nominal, but also at a Bonferroni-corrected level of significance ($P_{\text{WHR}} < 0.05/53$ for all variants). The novel **supplementary table 2** is cited at line 151:

Of note, all 53 WHRadjBMI-derived variants showed a significant WHR association in the UK Biobank data ($P_{\text{WHR}} < 3 \times 10^{-4}$, **Supplementary Table 2**).

+ Line 120. For WHRadjBMI, several loci have highly differentiated effects between males and females. Do female-specific loci always fall into the same category, and if they do, is the same true for the few male-specific loci?

The reviewer is right that sex-differences are an issue for WHRadjBMI loci. In our data, there are 11 loci with significant sex-differences among the 53 WHRadjBMI-derived variants ($P_{\text{Sexdiff}} < 0.05/53$; 10 women-specific; one men-specific, **Supplementary Table 3**). Indeed, the women-specific loci fall into the same category when conducting the classification by their women-specific co-association on BMI and WHR (see **Supplementary Table 3** for the result of the sex specific re-classification). The one men-specific variant remains in the WHRonly- class when the classification is based on men-specific estimates. We agree with the reviewer that this was not fully clear in the previous version of the manuscript. We have thus extended the text in the manuscript on the sensitivity analyses as follows and hope that this is now clearer (line 154):

Since WHRadjBMI is known for sexually dimorphic genetic effects^{8,9}, we also conducted a sensitivity analysis re-classifying the variants based on their sex-specific association (i.e. classifying variants based on their effect on WHR and BMI in women or their effect in men). While most of the 159 variants retained their initial class, the 11 WHRadjBMI-derived variants with significant sex-difference in our data ($P_{\text{Sexdiff}} < 0.05/53$) retained their initial class when the classification was performed on the sex with the larger effect, i.e., on women-specific associations for the 10 women-specific loci, or on men-specific associations for one men-specific locus (**Supplementary Table 3**).

+ Line 170. I found this paragraph very difficult to follow. I think it would be worth adding a couple of sentences about the previous study that attempted to identify/classify favourable adiposity loci. Maybe this would be better removed from the results, and just left in the discussion.

We agree with the reviewer that this was a difficult part to follow. We have moved this section to the discussion and re-worded to make it clearer, as advised (line 357):

There have been different approaches to capture favorable adiposity. Among the 11²⁴ or 53²⁸ loci previously identified for insulin resistance and put into context with favorable adiposity, 7 or 13 loci, respectively, capture favorable adiposity effects in the here utilized data following a definition where the BMI-increasing allele ($P_{\text{BMI}} < 0.05$) shows decreased risk of T2D or CAD (P_{T2D} or $P_{\text{CAD}} < 0.05$, no increased risk in either). To be comparable, we derived 1 Mb regions around our 159 lead variants resulting in 117 distinct regions. Of these 117 regions, excluding *TCF7L2* due to its extreme T2D risk (and potential index event bias in the BMI-association^{5,35}), 16 regions contained one of our 159 signal variants with a favorable adiposity effect. Of these 16 regions, 10 were novel compared to previous

work^{24,28}, 10 were classified as *BMI+WHR-* including seven novel (**Supplementary Table 16**). We were thus able to increase the number of loci with favorable adiposity by 50%.

+ Line 184. Excluding Metabochip data – I was confused by this, and it was not clear how the data used for these analyses differed from the original GIANT studies used for classification (and why this was necessary).

We apologize for the confusion. The Metabochip data was removed because the Metabochip variant panel over-represents loci from metabolic parameters and thus is not systematic or objective with regard to the coverage. We did this in the same fashion as previously published in Shungin et al., Nature 2015, and Locke et al., Nature 2015. We have extended the text and added references to the previous studies as follows (line 238):

For this, we used DEPICT¹⁹ to search for enriched pathways among the genes overlapping association signals ($P < 10^{-5}$ for any of BMI, WHR, or WHRadjBMI, excluding metabochip data as done previously^{5,6}, to avoid enriching for known metabolic regions by chip design, **Online Methods**).

+ Line 199. I didn't understand what these different exclusions meant, and why they were being performed.

We now stated this more specifically to clarify. First, we conducted a sensitivity analysis excluding height loci variants. This was because the *WHRonly-* class variants include a strong genetic effect on height harboring the *GDF* gene and we wanted to make sure that our results are not driven by height effects. Second, we conducted a sensitivity analysis excluding the full *RSPO3* signal and another sensitivity analysis including only one signal for *RSPO3* by using a wider region definition. This was to make sure that the results were not only triggered by the *RSPO3* signal or by the multiple signals within genomic regions like *RSPO3*. We have extended the wording accordingly (line 255):

This *WHRonly-* class finding was robust, even more pronounced, after excluding known height loci (to remove effects of the known strong height locus around *GDF* and other height regions), after excluding all five *RSPO3* signals (to limit the strong contribution of multiple *RSPO3* signals in this class), or after using a wider locus definition treating the *RSPO3* signals as a single region in the DEPICT analyses (to limit the contribution of multiple signals like *RSPO3*, **Supplementary Figure 7, Supplementary Tables 11-13**).

+ Line 202. For the eQTL analyses, why use D-prime=1 as a threshold. The more relevant metric is r2. Furthermore, the fact that a variant is also an eQTL does not mean that the trait association signal and eQTL signal are coincidental.

We did use the D-prime=1 as threshold, as this would include variants that are inherited together with the identified association signal. However, we agree that r2 would be more appropriate. Based on this comment and comments on the biological follow up by the other reviewers, we have rethought our biological follow up strategy. We agree with all reviewers that this was a weakness of

the manuscript. We decided to fully remove our ad-hoc and manual GTEx lookups. Instead, we now include (for each class) biological follow up analyses of tissue specific expression effects (based on GTEx v6) using the novel FUMA web-tool that was published recently (PMID 29184056, <http://fuma.ctglab.nl/>). Interestingly, we found that only the WHRonly- class signals enrich for genes with expression effects in digestive tissues of GTEx (two significant enrichments, 'Colon_Sigmoid' and 'Esophagus_Gastroesophageal_Junction'). While tissues are not directly comparable between DEPICT and FUMA (based on GTEx), this pattern was consistent between the two biological follow-up tools. We have included the FUMA results in the text (line 262):

To follow-up this finding, we used FUMA²⁰ to examine data from GTEx²¹ for tissue-specific enrichments of expression effects of genes overlapping our association results ($P < 10^{-5}$ for any of BMI, WHR, or WHRadjBMI, excluding metabochip data), again separating the variants by class. Consistent with the class-specific DEPICT analyses, genes harbouring *WHRonly*- variants were significantly enriched (Bonferroni-adjusted $P < 0.05$) for expression effects in an adipocyte-related tissue ('Adipose_Subcutaneous') as well as in digestive tissues ('Colon_Sigmoid' and 'Esophagus_Gastroesophageal_Junction', **Figure 6B, Supplementary Table 14, Supplementary Figure 8**). We found an overlap of nine genes (*BARX1*, *FOXP2*, *HOXA13*, *LAMB1*, *PCK1*, *PPARG*, *RGMA*, *RSPO3*, and *VEGFA*) that contributed to the significant digestive system results in both DEPICT and FUMA tissue specificity analyses of *WHRonly*- class variants.

We have extended the previous main figure on the WHRonly class tissue specific enrichments from DEPICT by FUMA results:

Figure 6. Tissue-specific enrichment of gene expression for WHRonly- class variants: Shown are results of DEPICT and FUMA tissue-specificity analyses based on variants that were selected from GWAS-only meta-analyses of GIANT ($P < 10^{-5}$) and that were classified as WHRonly-. Significant results within the digestive system are marked with green arrows. **A:** DEPICT results for WHRonly- with significant enrichments highlighted in blue (FDR<5%). Results are grouped by type and ordered alphabetically by MeSH term within a specific system, cell type or tissue (details in **Supplementary Table 10**). Results for the other three classes showed no significance with DEPICT (**Supplementary Figure 6**). **B:** FUMA results with significant enrichments highlighted in red (adjusted $P < 0.05$, Bonferroni-corrected, details in **Supplementary Table 14**). The $-\log_{10}(P)$ values in the graph refer to the probability of the hypergeometric test. Results for the other three classes showed only little enrichment with FUMA (**Supplementary Figure 8**).

We have added full FUMA results on the tissue specificity to supplementary tables (new **Supplementary Table 14**) and include result figures for the three other classes in supplement (new **Supplementary Figure 8**). We also include a methods section for the FUMA analyses (line 581).

+ Line 212. The “wrap-up” paragraph would be better in the discussion.

We do agree that this paragraph is a summary of the results. However, given the large abundance of presented data, we feel that a meta-level of the results is important to obtain the overview, which we would not like so much to mingle with the discussion. Therefore, we would like to keep it as a “wrap-up” as a meta-level of results and hope this is OK.

+ Line 272. I didn’t understand where the various numbers of loci were coming from – please clarify.

We apologize for the confusion. These numbers referred to the overlap between BMI+WHR- class variants and loci to be defined as favorable adiposity loci. We have re-written the paragraph in Discussion as described in the answer to the previous comment (see above) and hope that this clarifies the issue (line 357).

Line 302. There is a point (c), but I couldn't see point (b)!

We apologize for the mistake. We have restructured the whole paragraph based on another reviewer comment so that this is corrected now.

#####

Reviewer #3 (Remarks to the Author):

The revised manuscript from Winkler and colleagues satisfactorily addresses some but not all reviewer comments. Specifically, the report better addresses one concern of Aschard et al, but the conclusion that this analysis identifies novel biology is still weak and not well supported by the data. Some changes to the manuscript raise new concerns.

1. The new analyses of UK Biobank data to address collider bias provide a valuable contribution to the literature, especially showing that analysis of WHRadjBMI did not generate 'spurious' associations with the adjusted trait, even with improved power. However, the text could still be improved to better describe this contribution.

We agree with the reviewer that this was an important finding that could be better described. Based on this comment and comments by other reviewers, we have added a novel **Supplementary Table 2** with BMI and WHR association results from the UK Biobank (N=336,107). All of the 53 WHRadjBMI-derived variants ($P_{\text{WHRadjBMI}} < 5 \times 10^{-8}$ in GIANT) are significantly associated with WHR in the UK Biobank data ($P_{\text{WHR}} < 0.05/53$ for all variants) precluding spurious associations for the 53 WHRadjBMI-derived variants in the sense that they all have an effect on WHR. The novel **supplementary table 2** is cited at line 151:

Of note, all 53 WHRadjBMI-derived variants showed a significant WHR association in the UK Biobank data ($P_{\text{WHR}} < 3 \times 10^{-4}$, **Supplementary Table 2**).

- Introduction lines 57-58 are need to more clearly explain the concern being addressed here

We agree with the reviewer to be more specific and clear on the concern in the introduction already. We have thus expanded the introduction to explain more clearly the concern raised by Aschard et al. (line 44):

Frequently, this is approached by using an adjusted model trait where the trait Y is adjusted for a covariate Z (YadjZ) in order to separate the genetics of YadjZ from the genetics of Z. However, these adjusted model traits warrant cautious interpretation: as Aschard and colleagues pointed out, genome scans for traits adjusted for heritable covariates reveal not only genetic factors for the phenotype Y, but also those of the covariate Z to an extent that depends on their correlation¹.

And also starting at line 59:

Aschard and colleagues pointed out that some of the WHRadjBMI lead variants were not completely independent of BMI and showed some effect on BMI in the unexpected direction (WHR increasing allele decreased BMI). This is due to the fact that the genetic effect estimate for WHRadjBMI, $b_{\text{WHRadjBMI}}$, is related to the estimate for WHR, b_{WHR} , and the estimate for BMI, b_{BMI} , by $b_{\text{WHRadjBMI}} = b_{\text{WHR}} - r * b_{\text{BMI}}$, with r being the observational correlation between BMI and WHR in the analyzed study¹. A genome-wide scan on WHRadjBMI will thus not only identify genetic factors for WHR, but will also tend to pick up variants with an additional opposite effect on BMI, or even an effect on BMI only when the sample size is large enough. Aschard and colleagues extended their point by cautioning against potentially false positive signals and biased genetic effect estimates. They propose to

examine the potential of the bias by investigating the corrected effect $b_{\text{WHRadjBMI}} + r * b_{\text{BMI}}$ to ensure that an established WHRadjBMI-association is not biased by the BMI-association.

- The new Discussion on pages 9-10 is quite difficult to follow and needs clarification. For example, lines 294-295, how do the data confirm that WHRadjBMI GWAS may include variants with only a BMI effect when no such examples are found? Separate the supporting results from opinion.

Also, line 296 has point (a) and line 302 has point (c), but there is no point (b). Consider further paragraph divisions to separate points, and further attention to clear sentence grammar and paragraph structure.

We agree with the reviewer that this discussion section was still a bit difficult and according this comment (and a similar comment by another reviewer), we have substantially restructured this section.

We now structured this section by first commenting on the Aschard points that are truly valid and important points and which we can confirm (line 368), and second, commenting on the misconceptions about the implications of Aschard et al points to which we can provide empirical data to clarify that they are misconceptions (line 377). Third, we comment on the aspect that WHRadjBMI GWAS was perceived as treacherous and less useful than a WHR GWAS as a consequence of these misinterpretations of Aschard et al points (line 406) and, forth we provide specific recommendations for adiposity genetics research (line 417). We find that this really improved the discussion and we thank the reviewer for motivating this.

2. Interpreting loci based on the direction of effect on both BMI and WHR, dividing each group in two, seems like it could be valuable. However, the additional analyses regarding digestive and urogenital systems, providing lists of gene names and some eQTL lookups without further interpretation, did little to support the conclusion that these systems play a role in obesity genetics or that this analysis detects novel biology. The lack of substantial new biological insights remains a limitation of the work.

Indeed the lack of substantiated new biology in terms of pathways and eQTL results could have been viewed as a weak point of the paper. However, we would like to point out a few key aspects.

1. Our biological follow-up is purely bioinformatic-based and no experiments have been performed to further support the involvement of these implicated systems. However, this is a general criticism that is valid to the vast majority of computational analysis / method papers. DEPICT tissue-enrichment analysis revealed that genes proximal to WHR-only associated SNPs have enriched expression levels in the digestive system. While we think that these results are interesting and worthwhile to point out, we agree with the reviewer that the confirmation of the finding in independent data was a weakness of the previous version of the manuscript. Based on the comment and similar comments by other reviewers, we now include (for each class) additional biological follow-up analyses of tissue specific expression effects using the novel FUMA web-tool that was published recently (PMID 29184056, <http://fuma.ctglab.nl/>), which utilizes GTEx v6. Consequently we

removed our ad-hoc and manual GTEx lookups. Interestingly, we found digestive system enrichments for the WHRonly- class variants that were supportive of the findings from DEPICT. We have added these results to the text (line 262), to the new main **Figure 6**, to the online methods (line 581) and to the supplementary material (new **Supplementary figure 8** and **Supplementary table 14**). We do find that this supports the bio-informatically detected potentially novel biology.

Still, we agree with the reviewer that further data and experiments will be necessary to determine the mechanism through which these variants can be linked to transcriptomic regulation in the digestive system. We have acknowledge this in the discussion now (see line 350):

Still, further data and experiments will be necessary to determine the mechanisms through which these variants can be linked to transcriptional regulation in digestive systems.

2. Furthermore, we would like to point out that there is substantial novelty in our results linking our classes of adiposity variants to differential effects on fat depots. We found very distinct patterns on of VAT, SAT, and VAT/SAT by class that are health relevant.

3. Finally, we substantially contribute to the ongoing discussion of favourable adiposity by showing that specifically variants in one class (BMI+WHR-) are linked to metabolic favourable profile and jointly decrease T2D as much as down to 10% (other direction: 10-fold increase of T2D risk). Additionally, we find that the BMI-increasing variants in this class are causally linked to increased hip circumference and SAT, while there is no effect on waist or VAT. This supports the notion that this beneficial BMI-increase is due to directly beneficial effects of increased hip (or hip SAT) and thus contributes to the ongoing discussion of whether hip fat is directly or indirectly (via avoiding fat depots on more detrimental places). We have felt that this might have been not fully clear in the previous version. We have thus expanded the description of these aspects (see lines 297, 339).

In the light of these, we feel that our findings from bioinformatical follow-up reveal important biologically and medically relevant aspects and substantially contribute evidence to ongoing discussions.

3. The description of WHRadjBMI as a measure of “unusual fat distribution” seems forced and too specific to extreme values. Analyzing any quantitative trait is not limited to revealing effects leading to unusual values but identifies effects leading to variation across the trait range. Here, usual is defined based trait correlation in all members of a population sample and unusual is based on an arbitrary p-value threshold. The analyses CAN DETECT unusual fat distribution, but the trait itself is not a measure of unusual fat distribution. The wording should be changed.

We agree with the reviewer that our wording alluded too much to extreme values. We re-worded to “WHRadjBMI as a measure of unexpected fat distribution given the BMI” (e.g. line 36) to be more specific on what WHRadjBMI measures. We have amended the wording throughout the manuscript and hope that this is now clearer.

Minor point: the loci listed on line 107 with reference to Figure 1 are slightly different names than presented in the Figure: ANKRD65 vs ANKRD55 and CACLRL vs CALCRL

We apologize for the confusion and have corrected this error in the text (line 128).

#####

Reviewer #4 (Remarks to the Author):

Winkler et al use previously published GWAS meta-analysis results to explore the effect of BMI on waist-hip ratio signals, to partially rebut the critiques on employing "adjusted model" traits. They then use these comparisons to categorize BMI and WHR loci based on their co-association (or lack thereof) with the two traits to identify distinct subsets among, and then to characterize the biological relationships among and between these sets of loci. They argue that this differentiation helps identify novel physiological systems not previously implicated in obesity biology, namely digestive and urogenital systems.

This analysis is an interesting approach to both understanding the complexities of "adjusted model" traits, and for teasing apart the complex biology of obesity. While the authors generate little data on their own, their approach in bringing these data together to both address concerns about statistical methodology, and for understanding the results are valuable, and helpful in understanding both aspects. While their conclusions on the lack of biased and thus spurious findings for WHRadjBMI can only be as convincing as the current sample data allow, the consistency of the GIANT results with UKBB are reassuring. The robustness of their DEPICT and expression results are less convincing, and could be strengthened to convince the reader that these effects are specific to the subtype and thus real.

We thank the reviewer for the excellent summary and we agree with the review on all accounts.

We agree that lack of robustness of DEPICT results and expression results was a weakness of the previous version of the manuscript. Based on this and comments raised by other reviewers, we have re-thought our strategy. We decided to exclude our previous manual ad-hoc GTEx analysis and instead now include class-specific analyses of tissue specific expression effects in GTEx v6 data using the novel biological follow-up tool FUMA that was published recently (PMID 29184056, <http://fuma.ctglab.nl/>). Interestingly, the results were supportive of DEPICT in a sense that we replicated digestive system enrichments for the WHRonly- class variants. We have added these results to the text (line 262), main **figure 6**, to the online methods (line 581) and to the supplementary material (new **supplementary figure 8** and **supplementary table 14**).

Comments:

1) It was imperative that the authors make a more explicit comment on the Aschard paper, and they make important points on the distinctions between how the Aschard findings have been interpreted and how these adjusted models function in reality. In particular, clarifying the past nomenclature of "independent from BMI" is important. And the additional mathematical justifications, though mostly in the supplement, are important justifications for the intuitive, but common misconceptions. It is understandable, but too bad that the comparisons of expected to calculated results could not be highlighted more to more solidly justify the rest of the results.

We thank the reviewer for this excellent feed-back. Based on the comment, we have decided to strengthen the scope of the paper on the Aschard et al points. We have extended on the aspect in introduction (lines 44ff, 59ff) and have restructured the discussion paragraph (line 368-425).

Specifically, we have added a new chapter in the results about the comparison of calculated and ‘estimated’ genetic effects (line 167), that was previously in the Supplementary Note:

Computing the WHR effect from observed BMI and WHRadjBMI effects

When $b_{WHRadjBMI}$ and b_{BMI} are given for a variant, b_{WHR} can be computed as $b_{WHRadjBMI} + r \cdot b_{BMI}$ (or $b_{WHRadjBMI}$ as $b_{WHR} - r \cdot b_{BMI}$). We aimed to provide empirical data of how good this computation works by comparing the b_{WHR} estimates computed as described above with the observed b_{WHR} (**Figure 3**). When conducting this comparison in one study where we could estimate r directly (interim UK Biobank, $N=116,295$, $r=0.44$), we found perfect agreement between computed and observed b_{WHR} (Spearman correlation coefficient >0.98). When conducting this comparison in a meta-analysis setting where r could not be estimated directly (i.e. in GIANT, using r from UK Biobank as a reasonable average across GIANT studies), we found still a strong agreement (Spearman correlation coefficient $=0.88$). We were able to improve this agreement even further by using sex-stratified correlation estimates (from UK Biobank, $r=0.46$ for women, 0.60 for men) and sex-stratified effect estimates (from GIANT, Spearman correlation coefficient $=0.95$). Therefore, the formula $b_{WHR} = b_{WHRadjBMI} + r \cdot b_{BMI}$ can very well be used to compute unadjusted estimates from adjusted estimates and BMI estimates; the corresponding standard errors are, however, slightly increased yielding lower power (**Supplementary Note 1, Supplementary Figure 2**). As a consequence, for consortia working with obesity traits, such as GIANT^{5,6}, the number of genome-wide traits to be modeled can be limited to two traits as the effect estimate from the third trait can be re-computed with a small loss in precision.

Appertaining to the novel results chapter, we have moved the respective figure from the supplement to the main document, novel main **Figure 3**.

We felt that the aspect was lacking, how the computed standard error versus the data-estimates standard error of the WHR effect relates to each other. Thus, we have added a new **supplementary note 1** and added a new **Supplementary Figure 2** on the comparison of standard errors for the calculated or estimated WHR effects (cited from text, line 182).

Supplementary Note 1. Computing variance estimates of WHR effects from meta-analyzed BMI and WHRadjBMI effects and their variance estimates.

Based on the effect for BMI and WHRadjBMI and their variance estimates, the WHR effect and its variance can be computed via $b_{WHR} = b_{WHRadjBMI} + r \cdot b_{BMI}$ and $\widehat{Var}(b_{WHR}) = \widehat{Var}(b_{WHRadjBMI}) + r^2 \widehat{Var}(b_{BMI}) + 2r \widehat{Cov}(b_{WHRadjBMI}, b_{BMI})$. Here, r denotes the phenotypic correlation between WHR and BMI in the study – or the average of all included studies in a meta-analysis setting. Under the assumption of a covariance between BMI and WHRadjBMI effect close to zero (Spearman correlation coefficient between $b_{WHRadjBMI}$ and b_{BMI} in GIANT data is <0.01), we can derive $\widehat{Var}(b_{WHR})$ as $\widehat{Var}(b_{WHR}) = \widehat{Var}(b_{WHRadjBMI}) + r^2 \widehat{Var}(b_{BMI})$. Since this computation of $\widehat{Var}(b_{WHR})$ involves the sampling error of both, the WHRadjBMI effect and the BMI effect, this computed variance estimate will be expected to be larger than the observed variance estimate. We exemplify how well this computation of $\widehat{Var}(b_{WHR})$ compared to the directly estimated $\widehat{Var}(b_{WHR})$ in practice. For the 38 genome-wide significant WHR variants from GIANT, the

WHR standard errors were re-calculated from BMI and WHRadjBMI GIANT meta-analysis estimates. These are on average 10.3% larger than the standard errors from the original WHR meta-analysis (Supplementary Figure 2).

Supplementary Figure 2. Comparison of estimated and computed WHR standard errors for the 38 genome-wide significant WHR-derived lead variants. Using GIANT meta-analysis summary statistics, we compare standard errors of meta-analysed overall WHR effects (resulting from meta-analysis of multiple studies) with standard errors of computed WHR effects that were calculated from meta-analysed overall BMI and WHRadjBMI effects using the overall correlation between WHR and BMI ($r=0.44$, in UKBB).

2) The authors have clearly taken pains to make the classification of each locus into their 4 categories carefully. However, it is still extremely difficult to follow in the text, and distributions in Figures 1 A and C don't help to clarify much.

We agree, that our previous effort to make clear which variants is derived from which scan and how these distribute across the classes was not yet fully successful. We have therefore re-structured previous Figure 1A,B,C to now a **Figure 1** and **Figure 2A,B** in the attempt to make this easier to comprehend.

- How are none of the BMI+WHR- loci associated with BMI? in the Venn Diagram

In the Venn Diagram, we did not look at nominal significant association with BMI, but at genome-wide significant association with BMI. Indeed, none of the BMI+WHR- variants showed genome-wide significant association with BMI (all of them show BMI association P-Values between 0.05 and 5×10^{-8}). This should now be clearer in the current **Figure 1**.

- Why do the numbers in parentheses in 1C not add up to the expected totals?

In previous Figure 1C, the numbers did not add up because there is an overlap in genome-wide significant variants between the three scans (which was shown in previous Figure 1A). We apologize for the confusion and have tried to clarify this in the new version of **Figures 1 and 2**.

In the new **Figure 1**, we now show the overlap between variants detected at genome-wide significance for any of the three scans (BMI, WHR or the WHRadjBMI, no classification introduced at this stage to prevent confusion). Accordingly, we re-worked the panels 1A and 1B to a new **Figure 2** showing the results of the classification that now also includes Venn diagrams per class to allow the reader an easy overview on the source of the respective variants. We feel that this has increased markedly the clarity.

In addition, we have added, to various places in the manuscript, whether “significance” refers to nominal or genome-wide alpha levels (for example, see line 88). We hope the changes make the distinction much clearer to the reviewer/reader.

See here the new Figure 1 and Figure 2 for your convenience:

Figure 1. Identification of 159 genome-wide significant lead variants from three genomic scans. The Venn diagram shows the number of independent genome-wide significant ($P < 5 \times 10^{-8}$) signals derived from the BMI-, the WHR-, or the WHRadjBMI-scan, respectively, and their overlap. We found no overlap between BMI- and WHRadjBMI-derived variants.

Figure 2. Classification of the 159 signals according to the position on the b_{WHR} - b_{BMI} -plane and their overlap by scan. A. The Scatter plot shows the 159 variants on the b_{WHR} - b_{BMI} -plane, where b_{WHR} and b_{BMI} are the variant’s effect on WHR and BMI, respectively. Coloring indicates the four classes: *BMI+WHR+* (blue, nominal significant effects on BMI and WHR with consistent directions), *BMIonly+* (green, nominal significant effects on BMI only), *WHRonly-* (purple, nominal significant effects on WHR only) and *BMI+WHR-* (red, nominal significant effects on BMI and WHR with opposite directions). Symbols indicate a nominal significance purely for BMI ($P_{BMI} < 0.05$, $P_{WHR} \geq 0.05$, downward triangle), purely for WHR ($P_{BMI} \geq 0.05$, $P_{WHR} < 0.05$, upward signals), or for both ($P_{BMI} < 0.05$, $P_{WHR} < 0.05$, stars). The dashed line indicates a null effect for WHRadjBMI ($b_{WHRadjBMI} = 0$, estimated as $b_{WHR} = r * b_{BMI}$, with the correlation between BMI and WHR estimated from the population-based CoLaus study,

$r=0.50$). **B:** The diagram shows the number of identified signals per class, illustrates the four classes in directed acyclic graphs and shows Venn diagrams per class to distinguish whether the signals were derived with genome-wide-significance by the BMI-, the WHR- or the WHRadjBMI-scan, or by multiple scans. The underlined numbers reflect the 53 genome-wide significant signals identified by the WHRadjBMI-scan.

3) I wonder if the choice of terminology is entirely appropriate. Why is one or another fat distribution "usual" or "unusual"? Is larger hip circumference (hip fat depots) really "favorable" or is it just relatively less bad? I worry that this may be somewhat misleading. Associations with shifts in SAT vs VAT are compelling, though.

We agree that the terminology is a point of potential debate. Due to this comment (and a similar comment by another reviewer), we re-worded our description of what WHRadjBMI measures now as the "expected WHR given the BMI" and a genetic effect on WHRadjBMI as an effect with a change in WHR that is unexpected given the change in BMI.

Indeed the reviewer's question on whether larger hip is really "favorable" or just "relatively less bad" is an important question of considerable debate in the community. We can contribute here by our class-specific Mendelian Randomization approach, where we show that the genetic variants of the BMI+WHR- class (instruments) increase hip and SAT, and do not alter VAT or waist, and that these variants are also linked to metabolic favorable profile (decreased T2D and CAD). According to the Mendelian Randomization approach using the genetic variants as instruments, this implies that the BMI-increase and hip-increase is causally linked to the metabolic favorable profile; since there is no effect on waist, this cannot be an effect from decreased waist (and thus less central body fat storage).

The point that larger hip circumference is "favorable" pertains only to the subtype of effects that derive from the BMI+WHR- class; this is not true for other subtypes, like those of the BMI+WHR+ effects as there a hip-increase goes together with a waist-increase and show a metabolically unfavorable adiposity.

This is an important point that we felt we needed to clarify. We have thus expanded this aspect at several places in the manuscript. In the abstract (line 32) , results (line 201, 297), and in the discussion (line 339).

Abstract:

Class-specific Mendelian randomization reveals that variants with simultaneous WHR-decrease and BMI-increase are linked to metabolically favorable adiposity through beneficial hip fat.

Results lines 201:

The class-specific view on the variants' co-association on hip and waist circumference revealed that *BMI+WHR+* and *BMIonly+* variants were hip and waist increasing, *WHRonly-* variants were enriched for hip increase and waist decrease, and the *BMI+WHR-* variants were enriched for hip-increasing effects that lacked effects on waist circumference (**Table 1**).

Results lines 297:

BMI+WHR- alleles increased hip and SAT, but had no effect on waist or VAT, and a markedly favorable metabolic profile (*metabolically favorable adiposity*, e.g. *GRB14-COBL1*). Our Mendelian Randomization approach²⁷ restricting the instruments to the *BMI+WHR-* variants showed that their BMI increasing effect was causally linked to a favorable metabolic profile, particularly decreased risk of T2D and CAD. We also showed that the BMI increase of *BMI+WHR-* variants was causally linked to increased hip circumference and SAT, but had no effect on waist circumference or VAT. This would be in line with a direct beneficial effect of SAT stored on hip, possibly through adipokines¹², for this subtype of adiposity effects.

Discussion, line 339:

Since the *BMI+WHR-* variants show no effect on waist circumference, their metabolically beneficial effect can only stem from a directly favorable effect from hip increase, but not from a less detrimental storage compared to central body fat (then the other allele would be waist-increasing).

4) A few comments on DEPICT analyses:

- I don't think the results in Supplement Fig. 6 are as convincing of previous findings as stated. It appears no findings are significant after subdividing SNPs into groups.

We do agree with the reviewer that the findings in the other three classes are not convincing when considering statistical significance. However, we can observe a similarity in the pattern of *BMI+WHR+* and *BMIonly+* (focusing on nervous system) and the pattern of *BMI+WHR-* (connective tissue) with the pattern observed previously by Locke et al. and Shungin et al, respectively (re-done by us again in a similar fashion and depicted in Supp Figure 5). We agree that our wording "our class-specific DEPICT analyses yielded a CONFIRMATORY pattern for CNS and adipose tissue" was not ideal. We have thus reworded to the following (line 249):

Our class-specific DEPICT analyses yielded a pattern for CNS and adipose tissue that was similar to the pattern observed previously by Locke et al. and Shungin et al. for three of our four classes^{5,6} (**Supplementary Figure 6, Supplementary Table 10**).

We hope that this is acceptable to the reviewer.

- Is there a way to use the set of 159 loci to test the robustness of the new tissue/system findings, to help convince that they aren't an artifact of dividing up the data arbitrarily? Could you permute sets the same size as the WHRonly- group to show that discovering new systems/tissues/cells is specific and not just a function of arbitrary subsetting and thus a chance finding?

Robustness of the DEPICT findings are a good question. We have decided to include additional follow-up analyses of tissue specificity using FUMA, which is a different approach using independent data. By this, we found a confirmation of the digestive systems involved specifically with the WHRonly- variants (see novel main text line 262, novel **Figure 6B**).

5) Additionally, the GTEx result trying to support the digestive system is weak, and shows very little specificity for the set of variants in which those tissues were identified.

We agree, our previous GTEx results were weak in their support of the DEPICT results. We have now added the FUMA analyses, which uses the GTEx data in a more elaborate way. These results support the DEPICT results. Please see novel main text line 262, novel **Figure 6B and the novel Supplementary Figure 8**.

Reviewer #2 (Remarks to the Author):

The authors have addressed my comments well. My only remaining (minor) comment relates to my previous comment about text on line 115. The authors should clarify that "nominally significant" is $p < 0.05$, and later should highlight that $p < 3 \times 10^{-4}$ is a Bonferroni correction, otherwise it is not clear where this threshold comes from.

Reviewer #3 (Remarks to the Author):

The reviewers have addressed my concerns. The more direct description of the concerns raised by Aschard et al and the revised discussion both improve this paper.

Reviewer #4 (Remarks to the Author):

The authors work to continue to clarify and improve the manuscript have improved its analyses, interpretations, and the readability. They have explain and demonstrate the concepts and interpretations of the theory at issue, and also present relevant and important dissection of obesity genetics. While a few of their conclusions remain less than convincing, they attempt to provide multiple lines of evidence to support their claims in a careful manner. The work adds thoughtful and intriguing nuance to obesity genetics, and it will be interesting to see similar relationships assessed as more data become available (such as those in the very recent Yengo Biorxiv paper).

Comments:

1) The additional comments on the Aschard paper in the introduction are good. They give a better indication of what could be expected in the coming applied analyses.

2) The new figures 1 and 2 are simpler and thus much clearer than the previous figure 1. Figure 2 legend doesn't seem quite right, though. Downward triangles should be $PWHR < 0.05$, $PBMI > 0.05$, Upward triangle for $PBMI < 0.05$, $PWHR > 0.05$. (and a small typo in the second to last sentence (thw should be the)).

3) The sentence on sex-specific classification is an important point to make, particularly for the strongly sex-influenced trait of WHR, but it reads awkwardly.

4) Were the digestive and urogenital signals statistically significant or just suggestive in DEPICT? They appear significant in the figure, could make this more clear in the text.

5) Was there any evidence of the "urogenital" signal in the FUMA analyses? (Doesn't look like it.) Are the "urogenital" tissue signals from DEPICT likely to be indicators of sex-specific effects or something else?

6) The added FUMA analyses add some support (consistency) to the DEPICT analyses, helping to bolster the credibility of this novel biology. Why isn't the nervous (tibial?) association mentioned? Inconsistencies between the analyses, such as the urogenital and nerve findings, which are ignored in the text, highlight the uncertainty and validity of these results.

7) The updated analyses, interpretation, and discussion of "metabolically-favorable adiposity" help to improve the understanding of this aspect of obesity biology. The additional fat associations and MR analyses help to further the authors interpretation, but I still think calling them truly "metabolically favorable" goes too far.

minor issues:

line 120: typo - missing semi-colon

line 152: is Pwhr a typo? Shouldn't it be $Pwhr < 3 \times 10^{-4}$?

Point-by-point response to Reviewers

#####

Reviewer #2 (Remarks to the Author):

The authors have addressed my comments well. My only remaining (minor) comment relates to my previous comment about text on line 115. The authors should clarify that "nominally significant" is $p < 0.05$, and later should highlight that $p < 3 \times 10^{-4}$ is a Bonferroni correction, otherwise it is not clear where this threshold comes from.

We thank the reviewer for the positive feedback. We have clarified the description of the significance thresholds as follows:

Line 140: "All 53 WHRadjBMI-derived variants had nominally significant effects on WHR ($P_{\text{WHR}} < 0.05$, i.e. no spurious associations, weakest WHR association observed in GIANT $P_{\text{WHR}} = 7.5 \times 10^{-3}$, **Supplementary Data 1**)."

Line 149: "A more stringent threshold at $P < 3 \times 10^{-4}$ ($= 0.05/159$, Bonferroni-corrected) resulted in 36 of the 53 WHRadjBMI-derived variants retaining the class, 11 variants changing from *BMI+WHR-* to *WHRonly-*, and six just missing the $P_{\text{WHR}} < 3 \times 10^{-4}$ in the GIANT data (one with BMI effect $P_{\text{BMI}} < 3 \times 10^{-4}$, five without any effect)."

#####

Reviewer #3 (Remarks to the Author):

The reviewers have addressed my concerns. The more direct description of the concerns raised by Aschard et al and the revised discussion both improve this paper.

We thank the reviewer for the positive feedback.

#####

Reviewer #4 (Remarks to the Author):

The authors work to continue to clarify and improve the manuscript have improved its analyses, interpretations, and the readability. They have explain and demonstrate the concepts and interpretations of the theory at issue, and also present relevant and important dissection of obesity genetics. While a few of their conclusions remain less than convincing, they attempt to provide multiple lines of evidence to support their claims in a careful manner. The work adds thoughtful and intriguing nuance to obesity genetics, and it will be interesting to see similar relationships assessed as more data become available (such as those in the very recent Yengo Biorxiv paper).

We thank the reviewer for the positive feedback.

Comments:

1) The additional comments on the Aschard paper in the introduction are good. They give a better indication of what could be expected in the coming applied analyses.

We thank the reviewer for the positive feedback.

2) The new figures 1 and 2 are simpler and thus much clearer than the previous figure 1. Figure 2 legend doesn't seem quite right, though. Downward triangles should be $P_{WHR} < 0.05$, $P_{BMI} > 0.05$, Upward triangle for $P_{BMI} < 0.05$, $P_{WHR} > 0.05$. (and a small typo in the second to last sentence (thw should be the)).

We thank the reviewer for the positive feedback. Indeed, the labelling was incorrect: downward triangles indicate WHR-only, upward triangles indicate BMI-only signals.

3) The sentence on sex-specific classification is an important point to make, particularly for the strongly sex-influenced trait of WHR, but it reads awkwardly.

We agree with the reviewer that the sentence was awkward. We have tried to simplify the presentation by focusing on the re-classification for their sex-specific association to the 53 variants from the WHRadjBMI variants, for which the substantial sex-influence was reported in previous work (Shungin et al., Nature 2015; Winkler et al., Plos Genet 2015). By this, we could substantially simplify the text and hope that the reviewer will agree:

This was the previous sentence:

“Since WHRadjBMI is known for sexually dimorphic genetic effects^{8,9}, we also conducted a sensitivity analysis re-classifying the variants based on their sex-specific association (i.e. classifying variants based on their effect on WHR and BMI in women or their effect in men). While most of the 159 variants retained their initial class, the 11 WHRadjBMI-derived variants with significant sex-difference in our data ($P_{\text{Sexdiff}} < 0.05/53$) retained their initial class when the classification was performed on the sex with the larger effect, i.e., on women-specific associations for the 10 women-specific loci, or on men-specific associations for one men-specific locus (**Supplementary Table 3**).”

We have reworded the sentence as follows:

“Second, since WHRadjBMI is known for sexually dimorphic genetic effects^{8,9}, we also conducted a sensitivity analysis re-classifying the 53 WHRadjBMI variants based on their sex-specific effects on WHR and BMI (i.e. women-specific or men-specific classification). Among those, 11 variants showed significant sex-difference in the genetic effect on WHRadjBMI in our data ($P_{\text{Sexdiff}} < 0.05/53$). Among those, the 10 variants with women-specific effects retained class in the women-specific, but not in the men-specific classification; similarly, the one variant with men-specific effect retained class in the men-specific, but not in the women-specific classification. For all other variants there was no remarkable pattern by the re-classification for sex-specific effects (**Supplementary Data 3**).”

4) Were the digestive and urogenital signals statistically significant or just suggestive in DEPICT? They appear significant in the figure, could make this more clear in the text.

Yes, they were significant in Depict analyses. We have clarified this in the text as follows:

"*WHRonly*- variants were not only significantly enriched (at FDR < 5%) for adipocyte-related cells and tissues as reported previously⁶, but also in physiological systems labeled 'digestive' (rectum, cecum, upper GI, esophagus, stomach) and 'urogenital' (genitals, uterus, endometrium, myometrium) (**Fig. 6a, Supplementary Data 9**)."

5) Was there any evidence of the "urogenital" signal in the FUMA analyses? (Doesn't look like it.) Are the "urogenital" tissue signals from DEPICT likely to be indicators of sex-specific effects or something else?

This is correct, there was no evidence for enrichment of expression effects for the *WHRonly* class variants in the FUMA analyses. We have clarified this in the text as follows:

"In contrast to DEPICT analyses, there was no significant enrichment for expression effects in urogenital tissue in FUMA analyses; there was an additional significant finding for 'tibial nerve' in FUMA, which is a tissue not included in DEPICT. "

It would indeed be interesting to evaluate whether the genes contributing to the observed enrichment of expression effects in urogenital tissues indicated by Depict are linked to the observed sexually dimorphic genetic effects on adiposity traits. We plan to follow up on this interesting aspect but we feel that this would be more appropriate when looking on sex-specific expression. As soon as sex-specific GTEx summary statistics will become publically available, we will follow up on this. Since our paper is already very comprehensive and includes a lot of information and discussion, we feel that expanding on this point of sex-specific genetic effects, sex-specific expression, and sex-specific tissue involvement, though being of interest, is beyond the scope of the current paper.

6) The added FUMA analyses add some support (consistency) to the DEPICT analyses, helping to bolster the credibility of this novel biology. Why isn't the nervous (tibial?) association mentioned? Inconsistencies between the analyses, such as the urogenital and nerve findings, which are ignored in the text, highlight the uncertainty and validity of these results.

This is an interesting point. Indeed the consistency between FUMA and DEPICT (and other packages) would require a more detailed and comprehensive analysis. We here contribute some aspects of a comparison but a larger and comprehensive investigation is warranted (beyond the scope of this work). We have extended the description of inconsistencies accordingly; please see our response to the previous comment 5.

7) The updated analyses, interpretation, and discussion of "metabolically-favorable adiposity" help to improve the understanding of this aspect of obesity biology. The additional fat associations and MR analyses help to further the authors interpretation, but I still think calling them truly "metabolically favorable" goes too far.

We understand that the link between adiposity and a favorable profile is not fully straightforward as one would always link adiposity with an unfavorable profile. We now use a more cautious term "metabolically rather favorable adiposity" and hope that this will be acceptable.

minor issues:

line 120: typo - missing semi-colon

line 152: is Pwhr a typo? Shouldn't it be $P_{whr} < 3 \times 10^{-4}$?

We have corrected the typos – thank you!